# INSTANTEMBEDDING:
# EFFICIENT LOCAL NODE REPRESENTATIONS

## ABSTRACT

In this paper, we introduce InstantEmbedding, an efficient method for generating single-node representations using local PageRank computations. We theoretically prove that our approach produces globally consistent representations in sublinear time. We demonstrate this empirically by conducting extensive experiments on real-world datasets with over a billion edges. Our experiments confirm that Instant-Embedding requires drastically less computation time (over 9,000 times faster) and less memory (by over 8,000 times) to produce a single node's embedding than traditional methods including DeepWalk, node2vec, VERSE, and FastRP. We also show that our method produces high quality representations, demonstrating results that meet or exceed the state of the art for unsupervised representation learning on tasks like node classification and link prediction.

## 1 INTRODUCTION

Graphs are widely used to represent data when are objects connected to each other, such as social networks, chemical molecules, and knowledge graphs. A widely used approach in dealing with graphs is learning compact representations of graphs (Perozzi et al., 2014; Grover & Leskovec, 2016; Abu-El-Haija et al., 2018), which learns a $d$-dimensional embedding vector for each node in a given graph. Unsupervised embeddings in particular have shown improvements in many downstream machine learning tasks, such as visualization (Maaten & Hinton, 2008), node classification (Perozzi et al., 2014) and link prediction (Abu-El-Haija et al., 2018). Importantly, since such embeddings are learned solely from the structure of the graph, they can be used across multiple tasks and applications.

Typically, graph embedding models often assume that graph data fits in memory (Perozzi et al., 2014) and require representations for all nodes to be generated. However, in many real-world applications, it is often the case that graph data is large but also scarcely annotated. For example, the Friendster social graph (Yang & Leskovec, 2015) has only 30% nodes assigned to a community, from its total 65M entries. At the same time, many applications of graph embeddings such as classifying a data item only require one current representation for the item itself, and eventually representations of labeled nodes. Therefore, computing a full graph embedding is at worst infeasible and at best inefficient.

These observations motivate the problem which we study in this paper – the *Local Node Embedding* problem. In this setting, the embedding for a node is restricted to using only local structural information, and can not access the representations of other nodes in the graph or rely on trained global model state. In addition, we require that a local method needs to produce embeddings which are consistent with all other node's representations, so that the final representations can be used in the same downstream tasks that graph embeddings have proved adapt at in the past.

In this work, we introduce InstantEmbedding, an efficient method to generate local node embeddings *on the fly* in sublinear time which are globally consistent. Considering previous work that links embedding learning methods to matrix factorization (Tsitsulin et al., 2018; Qiu et al., 2018), our method leverages a high-order similarity matrix based on Personalized PageRank (PPR) as foundations on which local node embeddings are computed via hashing. We offer theoretical guarantees on the locality of the computation, as well as the proof of the global consistency of the generated embeddings. We show empirically that our method is able to produce high-quality representations on par with state of the art methods, with efficiency several orders of magnitude better in clock time and memory consumption: running 9,000 times faster and using 8,000 times less memory on the largest graphs that contenders can process.

Table 1: Related work in terms of desirable properties and the computational complexity necessary to generate a single node embedding. Note that all existing methods must generate a full graph embedding, and thus are directly dependent on the total graph size, while our method can directly solve this task in sublinear time. Analysis in Section 3.2.1.

| *method* | **Properties** | | | | **Complexities** | |
|---|---|---|---|---|---|---|
| | Local Inference | No Global Training | Unsupervised Embedding | Attribute-Free | Time $\mathcal{O}$ | Memory $\mathcal{O}$ |
| DeepWalk | ✗ | ✗ | ✔ | ✔ | $dn \log n$ | $dn + m$ |
| node2vec | ✗ | ✗ | ✔ | ✔ | $dbn$ | $n^3$ |
| VERSE | ✗ | ✗ | ✔ | ✔ | $dbn$ | $dn + m$ |
| FastRP | ✗ | ✗ | ✔ | ✔ | $dm\sqrt{n}$ | $dn + m$ |
| GCN | ✗ | ✗ | ✗ | ✗ | $dm$ | $dn + m$ |
| DGI | ✗ | ✗ | ✔ | ✗ | $dm$ | $dn + m$ |
| InstantEmbedding | ✔ | ✔ | ✔ | ✔ | $\frac{1}{\alpha(1-\alpha)\epsilon} + d$ | $\frac{1}{\alpha(1-\alpha)\epsilon} + d$ |

## 2 PRELIMINARIES & RELATED WORK

### 2.1 GRAPH EMBEDDING

Let $G = (V, E)$ represent an unweighted graph, which contains a set of nodes $V$, $|V| = n$, and edges $E \subseteq (V \times V)$, $|E| = m$. A graph can also be represented as an adjacency matrix $\mathbf{A} \in \{0, 1\}^{n \times n}$ where $\mathbf{A}_{u,v} = 1$ iff $(u, v) \in E$. The task of graph embedding then, is to learn a $d$-dimensional node embedding matrix $\mathbf{X} \in \mathbb{R}^{n \times d}$ where $\mathbf{X}_v$ serves as the embedding for any node $v \in V$. We note that $d \ll n$, i.e. the learned representations are low-dimensional, and the challenge of graph embedding is to best preserve graph properties (such as node similarities) in this space. Following the formalization in Abu-El-Haija et al. (2018), many graph embedding can be thought of minimizing an objective in the general form: $\min_{\mathbf{X}} L(f(\mathbf{X}), g(\mathbf{A}))$, where $f : \mathbb{R}^{n \times d} \to \mathbb{R}^{n \times n}$ is a *pairwise* distance function on the embedding space, $g : \mathbb{R}^{n \times n} \to \mathbb{R}^{n \times n}$ is a distance function on the (possibly transformed) adjacency matrix, and $L$ is a loss function over all $(u, v) \in (V \times V)$ pairs.

A number of graph embedding methods have been proposed. One family of these methods simply learn $\mathbf{X}$ as a lookup dictionary of embeddings and calculate the loss via distance (Kruskal, 1964), or matrix factorization (either implicit (Perozzi et al., 2014; Grover & Leskovec, 2016) or explicit (Ou et al., 2016)). Another line of work focuses on leveraging the graph structure using neighborhood aggregation (Battaglia et al., 2016; Scarselli et al., 2008), or the Laplacian matrix of the graph (Kipf & Welling, 2016). On attributed structured data, Graph Convolutional Networks (Kipf & Welling, 2016) have been successfully applied to both supervised and unsupervised tasks (Veličković et al., 2018). However, in the absence of node-level features, Duong et al. (2019) demonstrated that these methods do not produce meaningful representations.

**Graph Embedding via Random Projection** The computational efficiency brought by advances in random projection (Achlioptas, 2003; Dasgupta et al., 2010) paved the way for its adaptation in graph embedding to allow direct construction of the embedding matrix $\mathbf{X}$. Two recent works, RandNE (Zhang et al., 2018) and FastRP (Chen et al., 2019) iteratively project the adjacency matrix to simulate the higher-order interactions between nodes. As we show in the experiments, these methods suffer from high memory requirements and are not always competitive with other methods.

### 2.2 LOCAL ALGORITHMS ON GRAPHS

Local algorithms on graphs (Suomela, 2013) solve graph without using the full graph. A well-studied problem in this space is personalized recommendation (Jeh & Widom, 2003) where users are represented as nodes in a graph and the goal is to recommend items to specific users leveraging the graph structure. Classic solutions to this problem are Personalized PageRank (Gupta et al., 2013) and Collaborative Filtering (Schafer et al., 2007; He et al., 2017). Interestingly, these methods have been recently applied to graph neural networks (Klicpera et al., 2019; He et al., 2020). We now recall the definition of Personalized PageRank that is one of the main ingredients in our embedding algorithm.

**Definition** (Personalized PageRank (PPR, variation)). *Given* $\mathbf{s} \in \mathbb{R}^n$ *(*$\mathbf{s}_i \geq 0, \sum_i \mathbf{s}_i = 1$*), a distribution of the starting node of random walks, and* $\alpha \in (0,1)$*, a decay factor, the Personalized PageRank vector* $\pi(\mathbf{s}) \in \mathbb{R}^n$ *is defined recursively as:*

$$\pi(\mathbf{s}) = \alpha\mathbf{s} + (1-\alpha)\pi(\mathbf{s})^\top \frac{1}{2}(I + \mathbf{D}^{-1}\mathbf{A}), \tag{1}$$

*where* $\frac{1}{2}(I + \mathbf{D}^{-1}\mathbf{A})$ *is the lazy random-walk matrix.*

PPR takes as input a distribution of starting nodes $\mathbf{s}$, which is typically a $n$ dimensional one-hot vector $\mathbf{e}_i$ with 1 in the $i$-th coordinate, enforcing a local random walks starting from node $i$. Following this practice, we denote $\pi_i \in \mathbb{R}^n$, the PPR vector starting from a single node $i$, and $\mathbf{PPR} \in \mathbb{R}^{n \times n}$, the full PPR matrix for all nodes in the graph, where $\mathbf{PPR}_{i,:} = \pi(\mathbf{e}_i)$. VERSE (Tsitsulin et al., 2018) proposes to learn node embeddings by implicitly factorizing $\mathbf{PPR}$. Its stochastic approach can perform well, but lacks guarantees of stability and convergence. The idea of learning embeddings based on local random walks has also been used in the property testing framework, a direction in graph algorithm aiming at analyzing the clustering structure of a graph (Kale & Seshadhri, 2008; Czumaj & Sohler, 2010; Czumaj et al., 2015; Chiplunkar et al., 2018).

## 2.3 PROBLEM STATEMENT

In this work, we consider the problem of embedding a single node in a graph quickly. More formally, we consider what we term the *Local Node Embedding* problem: given a graph $G$ and any node $v$, return a globally consistent structural representation for $v$ using only local information around $v$, in time sublinear to the size of the graph.

A solution to the local node embedding problem should possess the following properties:

1. **Locality**. The embeddings for a node are computed locally, i.e. the embedding for a node can be produced using only local information and in time independent of the total graph size.
2. **Global Consistency**. A local method must produce embeddings that are globally consistent (i.e. able to relate each embedding to each other, s.t. distances in the space preserve proximity).

While many node embedding approaches have been proposed (Chen et al., 2018), to the best of our knowledge we are the first to examine the local embedding problem. Furthermore, no existing methods for positional representations of nodes meet these requirements. We briefly discuss these requirements in detail below, and put the related work in terms of these properties in Table 1.

**Locality.** While classic node embedding methods, such as DeepWalk (Perozzi et al., 2014), node2vec (Grover & Leskovec, 2016), or VERSE (Tsitsulin et al., 2018) rely on information aggregated from local subgraphs (e.g. sampled by a random walk), they do not meet our locality requirement. Specifically, they also require the representations of all the nodes around them, resulting in a dependency on information from all nodes in the graph (in addition to space complexity $O(nd)$ where $d$ is the embedding dimension) to compute a single representation. Classical random-projection based methods also require access to the full adjacency matrix in order to compute the higher-order ranking matrix. We briefly remark that even methods capable of local attributed subgraph embedding (such as GCN or DGI) also do not meet this definition of locality, as they require a global training phase to calibrate their graph pooling functions.

**Global Consistency**. This property allows embeddings produced by local node embedding to be used together, perhaps as features in a model. While existing methods for node embeddings are global ones which implicitly have global consistency, this property is not trivial for a local method to achieve. Specifically, a local method must produce a node representation that resides in a space, preserving proximities to all other node embeddings that may be generated, without relying on a global state. One exciting implication of a local method which is globally consistent is that it can wait to compute a representation until it is actually required for a task. For example, in a production system, one might only produce representations for immediate classification when they are requested.

We propose our approach satisfying these properties in Section 3, and experimentally validate it in Section 4, followed by conclusions in Section 5.

## 3 METHOD

Here we outline our proposed approach for local node embedding. We begin by discussing the connection between a recent embedding approach and matrix factorization. Then using this analysis, we propose an embedding method based on randomly hashing the PPR matrix. We note that this approach has a tantalizing property – it can be decomposed into entirely local operations per node. With this observation in hand, we present our solution, InstantEmbedding. Finally, we analyze the algorithmic complexity of our approach, showing that it is both a local algorithm (which runs in time sublinear to the size of $G$) and that the local representations are globally consistent.

### 3.1 GLOBAL EMBEDDING USING PPR

A recently proposed method for node embedding, VERSE (Tsitsulin et al., 2018), learns node embeddings using a neural network which encodes Personalized PageRank similarities. Their objective function, in the form of Noise Contrastive Estimation (NCE) (Gutmann & Hyvärinen, 2010), is:

$$\mathcal{L} = \sum_{i=1}^{n} \sum_{j=1}^{n} \left[ \mathbf{PPR}_{ij} \log \sigma \left( \mathbf{x}_i^\top \mathbf{x}_j \right) + b \mathbb{E}_{j' \sim \mathcal{U}} \log \sigma \left( -\mathbf{x}_i^\top \mathbf{x}_{j'} \right) \right], \tag{2}$$

where $\mathbf{PPR}$ is the Personalized PageRank matrix, $\sigma$ is the sigmoid function, $b$ is the number of negative samples, and $\mathcal{U}$ is a uniform noise distribution from which negative samples are drawn. Like many SkipGram-style methods (Mikolov et al., 2013), this learning process can be linked to matrix factorization by the following lemma:

**Lemma 3.1** (Tsitsulin et al. (2020)). *VERSE implicitly factorizes the matrix* $\log(\mathbf{PPR}) + \log n - \log b$ *into* $\mathbf{XX}^\top$*, where $n$ is the number of nodes in the graph and $b$ is the number of negative samples.*

#### 3.1.1 HASHING FOR GRAPH EMBEDDING

Lemma 3.1 provides an incentive to find an efficient alternative to factorizing the dense similarity matrix $\mathbf{M} = \log(\mathbf{PPR}) + \log n - \log b$. Our choice of the algorithm requires two important properties: a) providing an unbiased estimator for the inner product, and b) requiring less than $\mathcal{O}(n)$ memory. The first property is essential to ensure we have a good sketch of $\mathbf{M}$ for the embedding, while the second one keeps our complexity per node sublinear.

In order to meet both requirements we propose to use hashing (Weinberger et al., 2009) to preserve the essential similarities of $\mathbf{PPR}$ in expectation. We leverage two global hash functions $h_{\mathrm{d}} \colon \mathbb{N} \to \{0, ..., d - 1\}$ and $h_{\mathrm{sgn}} \colon \mathbb{N} \to \{-1, 1\}$ sampled from universal hash families $\mathbb{U}_d$ and $\mathbb{U}_{-1,1}$ respectively, to define the hashing kernel $H_{h_{\mathrm{d}}, h_{\mathrm{sgn}}} \colon \mathbb{R}^n \to \mathbb{R}^d$. Applied to an input vector $\mathbf{x}$, it yields $\mathbf{h} = H_{h_{\mathrm{d}}, h_{\mathrm{sgn}}}(\mathbf{x})$, where $\mathbf{h}_i = \sum_{k \in h_d^{-1}(i)} \mathbf{x}_k h_{\mathrm{sgn}}(k)$.

We note that although $H_{h_{\mathrm{d}}, h_{\mathrm{sgn}}}$ is proposed for vectors, it can be trivially extended to matrix $\mathbf{M}$ when applied to each row vector of that matrix, e.g. by defining $H_{h_{\mathrm{d}}, h_{\mathrm{sgn}}}(\mathbf{M})_{i,:} \equiv H_{h_{\mathrm{d}}, h_{\mathrm{sgn}}}(\mathbf{M}_{i,:})$. In the appendix we prove the next lemma that follows from (Weinberger et al., 2009) and highlights both the aforementioned properties:

**Lemma 3.2.** *The space complexity of $H_{h_{\mathrm{d}}, h_{\mathrm{sgn}}}$ is $\mathcal{O}(1)$ and:*

$$\mathbb{E}_{h_{\mathrm{d}} \sim \mathbb{U}_d, h_{\mathrm{sgn}} \sim \mathbb{U}_{-1,1}} \left[ H_{h_{\mathrm{d}}, h_{\mathrm{sgn}}}(\mathbf{M}) H_{h_{\mathrm{d}}, h_{\mathrm{sgn}}}(\mathbf{M})^\top \right] = \mathbf{MM}^\top$$

This matrix sketching technique is strongly related to the factorization proposed in Lemma 3.1. To better understand this, we consider the approximation $M \approx U\Sigma U^T$. If the (asymptotic) solution of VERSE is $U\sqrt{\Sigma}$, then our method aims to directly approximate $U\Sigma$. We show that this rescaled solution is more computationally tractable, while still preserving critical information for high-quality node representations.

Our algorithm for global node embedding is presented in Algorithm 1. First, we compute the PPR matrix $\mathbf{PPR}$ (Line 2) with a generic approach ($CreatePPRMatrix$), which takes a graph and $\epsilon$, the desired precision of the approximation. We remark that any of the many proposed approaches for computing such a matrix (e.g. from Jeh & Widom (2003); Andersen et al. (2007); Lofgren et al. (2014)) can be used for this calculation. As the $\mathbf{PPR}$ could be dense, the same could be said about the

implicit matrix $\mathbf{M}$. Thus, we filter the signal from non-significant $\mathbf{PPR}$ values by applying the $\max$ operator. We remove the constant $\log b$ from the implicit target matrix. In lines (4-6), the provided hash function accumulates each value in the corresponding embedding dimension.

---

**Algorithm 1** Global Node Embedding using Personalized PageRank

---

**Input:** graph $G$, embedding dimension $d$, PPR precision $\epsilon$, hash functions $h_d, h_{sgn}$
**Output:** embedding matrix $\mathbf{W}$
1: **function** GRAPHEMBEDDING($G, d, \epsilon, h_d, h_{\text{sgn}}$)
2: $\quad$ $\mathbf{PPR} \leftarrow CreatePPRMatrix(G, \epsilon)$
3: $\quad$ $\mathbf{W} = \mathbf{0}_{n \times d}$
4: $\quad$ **for** $\pi_i$ in $\mathbf{PPR}$ **do**
5: $\quad\quad$ **for** $r_j$ in $\pi_i$ **do**
6: $\quad\quad\quad$ $\mathbf{W}_{i, h_{\text{d}}(j)} \mathrel{+}= h_{\text{sgn}}(j) \times \max(\log(r_j * n), 0)$
7: $\quad$ **return** $\mathbf{W}$

---

Interestingly, the projection operation only uses information from each node's individual PPR vector $\pi_i$ to compute its representation. In the following section, we will show that local calculation of the PPR can be utilized to develop an entirely local algorithm for node embedding.

## 3.2 LOCAL NODE EMBEDDING VIA INSTANTEMBEDDING

Having a local projection method, all that we require is a procedure that can calculate the PPR vector for a node in time sublinear to size of the graph. Specifically, for InstantEmbedding we propose that the $CreatePPRMatrix$ operation consists of invoking the $SparsePPR$ routine from Andersen et al. (Andersen et al., 2007) once for each node $i$. This routine is an entirely local algorithm for efficiently constructing $\pi_i$, the PPR vector for node $i$, which offers strong guarantees. The next lemma follows from (Andersen et al., 2007) and formalizes the result (proof in Appendix A.4).

**Lemma 3.3.** *The* INSTANTEMBEDDING$(v, G, d, \epsilon)$ *algorithm computes the local embedding of a node $v$ by exploring at most the $O\left(1/(1-\alpha)\epsilon\right)$ nodes in the neighborhood of $v$.*

We present InstantEmbedding, our algorithm for local node embedding, in Algorithm 2. As we will show, it is a self-contained solution for the local node embedding problem that can generate embeddings for individual nodes extremely efficiently. Notably, per Lemma 3.3, the local area around $v$ explored by InstantEmbedding is independent of $n$. Therefore the algorithm is strictly local.

---

**Algorithm 2** InstantEmbedding

---

**Input:** node $v$, graph $G$, embedding dimension $d$, PPR precision $\epsilon$, hash functions $h_d, h_{sgn}$
**Output:** embedding vector $\mathbf{w}$
1: **function** INSTANTEMBEDDING($v, G, d, \epsilon, h_d, h_{sgn}$)
2: $\quad$ $\pi_v \leftarrow SparsePPR(v, G, \epsilon)$
3: $\quad$ $\mathbf{w} \leftarrow \mathbf{0}_d$
4: $\quad$ **for** $r_j$ in $\pi_v$ **do**
5: $\quad\quad$ $\mathbf{w}_{h_{\text{d}}(j)} \mathrel{+}= h_{\text{sgn}}(j) \times \max(\log(r_j * n), 0)$
6: $\quad$ **return** $\mathbf{w}$

---

### 3.2.1 ANALYSIS

We now prove some basic properties of our proposed approach. First, we show that the runtime of our algorithm is local and independent of $n$, the number of nodes in the graph. Then, we show that our local computations are globally consistent, i.e., the embedding of a node $v$ is the same independently if we compute it locally or if we recompute the embeddings for all nodes in the graph at the same time. Note that we focus on bounding the running time to compute the embedding for a *single* node in the graph. Nonetheless, the global complexity to compute all the embeddings can be obtained by multiplying our bound by $n$, although it is not the focus of this work. We state the following theorem and prove it in Appendix A.5.

**Theorem 3.4.** *The $InstantEmbedding(v, G, d, \epsilon)$ algorithm has running time $\mathcal{O}\left(d + 1/\alpha(1-\alpha)\epsilon\right)$.*

Besides the embedding size $d$, both the time and and space complexity of our algorithm depend only on the approximation factor $\epsilon$ and the decay factor $\alpha$. Both are independent of $n$, the size of the graph, and $m$, the size of the edge set. Notably, if $\mathcal{O}\left(1/\alpha(1-\alpha)\epsilon\right) \in o(n)$, as commonly happen in real world applications, our algorithm has sublinear time w.r.t. the graph size. Lastly, we note that the space complexity is also sublinear (due to Lemma 3.3), which we show in the appendix.

Now we turn our attention to the consistency of our algorithm, by showing that for a node $v$ the embeddings computed by $InstantEmbedding$ and $GraphEmbedding$ are identical. In the following we denote the graph embedding computed by $GraphEmbedding(G, d, \epsilon)$ for node $v$ by $GraphEmbedding(G, d, \epsilon)_v$, and we prove the following theorem (Appendix A.6).

**Theorem 3.5** (Global Consistency). *$InstantEmbedding(v, G, d, \epsilon)$ output equals one of $GraphEmbedding(G, d, \epsilon)$ at position $v$.*

**Complexity Comparison.** Table 1 compares the complexity of InstantEmbedding with that of previous works: $d$, $n$, $m$ stands for embedding dimension, size of graph and number of edges respectively. Specifically, $b \geq 1$ stands for the number of samples used in node2vec and VERSE. It is noteworthy that all the previous works have time complexity depending on $n$, and perform at least linear w.r.t. size of the graph. In contrast, our algorithm depends only on $\epsilon$ and $\alpha$, and has sublinear time w.r.t. $n$, the graph size. In Section 4, we experimentally verify the advantages of our principled method.

## 4 EXPERIMENTS

In the light of the theoretical guarantees about the proposed method, we perform extended experiments in order to verify our two main hypotheses:

1. **H1.** Computing local node-embedding is more efficient than generating a global embedding.
2. **H2.** The local representations are consistent and of high-quality, being competitive with and even surpassing state-of-the-art methods on several tasks.

We assess **H1** in Section 4.2, in which we measure the efficiency of generating a single node embedding for each method. Then in Section 4.3 we validate **H2** by comparing our method against the baselines on multiple datasets using tasks of node classification, link prediction and visualization.

### 4.1 DATASETS AND EXPERIMENTAL SETTINGS

To ensure a relevant and fair evaluation, we compare our method against multiple strong baselines, including Deep-Walk (Perozzi et al., 2014), node2vec (Grover & Leskovec, 2016), VERSE (Tsitsulin et al., 2018), and FastRP (Chen et al., 2019). Each method was run on a virtual machine hosted on the Google Cloud Platform, with a 2.3GHz 16-core CPU and 128GB of RAM. All reported results use dimensionality $d = 512$ for every method. We provide extended results for 4 additional baselines: RandNE (Zhang et al., 2018), NodeSketch (Yang et al., 2019), LouvainNE (Bhowmick et al., 2020) and FREDE (Tsitsulin et al., 2020) on a subset of tasks, along with full details regarding each

Table 2: Dataset attributes: size of vertices $|V|$, edges $|E|$, labeled vertices $|S|$.

| Dataset | $|V|$ | $|E|$ | $|S|$ |
|---|---|---|---|
| PPI | 3.8k | 38k | 3.8k |
| BlogCatalog | 10k | 334k | 10k |
| CoCit | 44k | 195k | 44k |
| CoAuthor | 52k | 356k | — |
| Flickr | 81k | 5.9M | 81k |
| YouTube | 1.1M | 3.0M | **32k** |
| Amazon2M | 2.4M | 62M | — |
| Orkut | 3.0M | 117M | **110k** |
| Friendster | 66M | 1806M | — |

method and its parameterization in the Appendix B.1. For reproducibility, we release an implementation of our method.[1]

**InstantEmbedding Instantiation**. As presented in Section 3, our implementation of the presented method relies on the choice of PPR approximation used. For instant single-node embeddings, we use the highly efficient PushFlow (Andersen et al., 2007) approximation that enables us to dynamically load into memory at most $2/(1-\alpha)\epsilon$ nodes from the full graph to compute a single PPR vector $\pi$. This is achieved by storing graphs in binarized compressed sparse row format that allows selective reads for nodes of interest. In the special case when a full graph embedding is requested, we have the freedom to approximate the PPR in a distributed manner (we omit this from runtime analysis, as

---

[1]Software available for reviewers and ACs in the OpenReview forum.

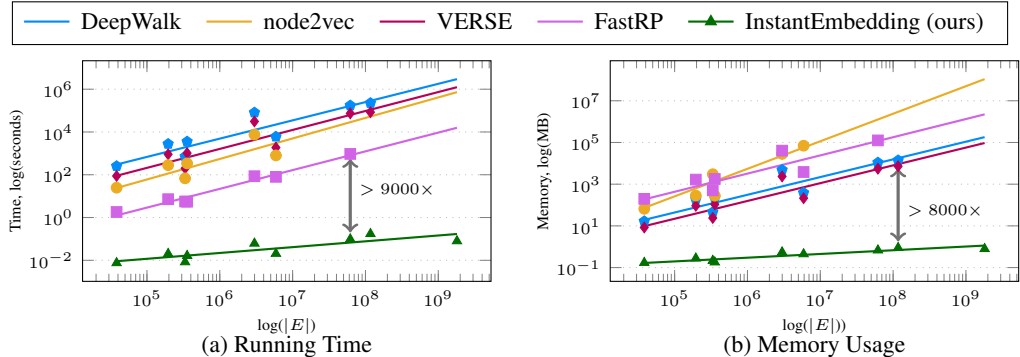

(a) Running Time          (b) Memory Usage

Figure 1: Required (a) running time and (b) memory consumption to generate a node embedding ($d$=512) based on the edge count of each graph ($|E|$), with the best line fit drawn. Our method is over **9,000** times faster than FastRP and uses over **8,000** times less memory than VERSE, the next most efficient baselines respectively, in the largest graph that these baseline methods can process.

we had no distributed implementations for the baselines, but we note our local method is trivially parallelizable). We refer to Appendix B.5 for the study of the influence of $\epsilon$ on runtime and quality.

**Datasets**. We perform our evaluations on 10 datasets as presented in Table 2. Detailed descriptions, scale-free and small-world measurements for these datasets are available in the supplementary material. Note that on YouTube and Orkut the number of labeled nodes is much smaller than the total. We observe this behavior in several real-world application scenarios, where our method shines the most.

## 4.2 PERFORMANCE CHARACTERISTICS

We report the mean wall time and total memory consumption (Wolff) required to generate an embedding ($d$=512) for a single node in the given dataset. Note that due to the nature of all considered baselines, they implicitly have to generate a full graph embedding in order to get a node representation. We repeat the experiment 1,000 times for InstantEmbedding due to its locality property; for the baselines, we average the running time from 5 experiments, and measure the memory usage once. For reference, we also provide the performance of InstantEmbedding producing a full graph embedding in Appendix B.3.

**Running Time**. As Figure 1(a) shows, InstantEmbedding is the most scalable method, drastically outperforming all the other methods, at the task of producing a single node embedding. We are over 9,000 times faster than the next fastest baseline in the largest graph both methods can process, and can scale to graphs of any size.

**Memory Consumption**.As Figure 1(b) shows, InstantEmbedding is the most efficient method having been able to run in all datasets using negligible memory compared to the other methods. Compared to the next most memory-efficient baseline (VERSE) we are over 8,000 times more efficient in the largest graph both methods can process.

The results of running time and memory analysis confirm hypothesis **H1** and show that Instant-Embedding has a significant speed and space advantage versus the baselines. The relative speedup continues to grow as the size of the datasets increase. On a dataset with over 1 billion edges (Friendster), we can compute a node embedding in 80ms – fast enough for a real-time application!

## 4.3 EMBEDDING QUALITY

**Node Classification.** This task measures the semantic information preserved by the embeddings by training a simple classifier on a small fraction of labeled representations. For each method, we perform three different random splits of the data. More details are available in the Appendix B.4.1.

In Table 3 we report the mean Micro F1 scores with their respective confidence intervals (corresponding Macro-F1 scores in the supplementary material). For each dataset, we perform Welch's

Table 3: Average Micro-F1 classification scores and confidence intervals. Our method is marked as follows: * - above baselines; **bold** - no other method is statistically significant better.

| Method \ Dataset | PPI | BlogCatalog | CoCit | Flickr | YouTube | Orkut |
|---|---|---|---|---|---|---|
| DeepWalk | $16.08 \pm 0.64$ | $32.48 \pm 0.35$ | $37.44 \pm 0.67$ | $31.22 \pm 0.38$ | $38.69 \pm 1.17$ | $87.67 \pm 0.23$ |
| node2vec | $15.03 \pm 3.18$ | $33.67 \pm 0.93$ | $38.35 \pm 1.75$ | $29.80 \pm 0.67$ | $36.02 \pm 2.01$ | DNC |
| VERSE | $12.59 \pm 2.54$ | $24.64 \pm 0.85$ | $38.22 \pm 1.34$ | $25.22 \pm 0.20$ | $36.74 \pm 1.05$ | $81.52 \pm 1.11$ |
| FastRP | $15.74 \pm 2.19$ | $33.54 \pm 0.96$ | $26.03 \pm 2.10$ | $29.85 \pm 0.26$ | $22.83 \pm 0.41$ | DNC |
| InstantEmbedding | $\mathbf{17.67}* \pm 1.22$ | $\mathbf{33.36} \pm 0.67$ | $\mathbf{39.95}* \pm 0.67$ | $30.43 \pm 0.79$ | $\mathbf{40.04}* \pm 0.97$ | $76.83 \pm 1.16$ |

Table 4: Average ROC-AUC scores and confidence intervals for the link prediction task. Our method is marked as follows: * - above baselines; **bold** - no other method is statistically significant better.

| Method \ Dataset | CoAuthor | Blogcatalog | Youtube | Amazon2M |
|---|---|---|---|---|
| DeepWalk | $88.43 \pm 1.08$ | $91.41 \pm 0.67$ | $82.17 \pm 1.02$ | $98.79 \pm 0.41$ |
| node2vec | $86.09 \pm 0.85$ | $92.18 \pm 0.12$ | $81.27 \pm 1.58$ | DNC |
| VERSE | $92.75 \pm 0.73$ | $93.42 \pm 0.35$ | $80.03 \pm 0.99$ | $99.67 \pm 0.18$ |
| FastRP | $82.19 \pm 2.22$ | $88.68 \pm 0.70$ | $76.30 \pm 1.46$ | $92.12 \pm 0.61$ |
| InstantEmbedding | $90.44 \pm 0.48$ | $92.74 \pm 0.60$ | $\mathbf{82.89}* \pm 0.83$ | $99.15 \pm 0.18$ |

t-test between our method and the best performing contender. We observe that InstantEmbedding is remarkably good on these node classification, despite its several approximations and locality restriction. Specifically, on four out of five datasets, no other method is statistically significant above ours, and three of these (PPI, CoCit and YouTube) we achieve the best classification results.

In Figure 2, we study how our hyperparameter, the PPR approximation error $\epsilon$, influences both the classification performance, running time, and memory consumption. There is a general sweet spot (around $\epsilon = 10^{-5}$) across datasets where InstantEmbedding outperforms competing methods while being orders of magnitude faster. Data on the other datasets is available in Section B.5.

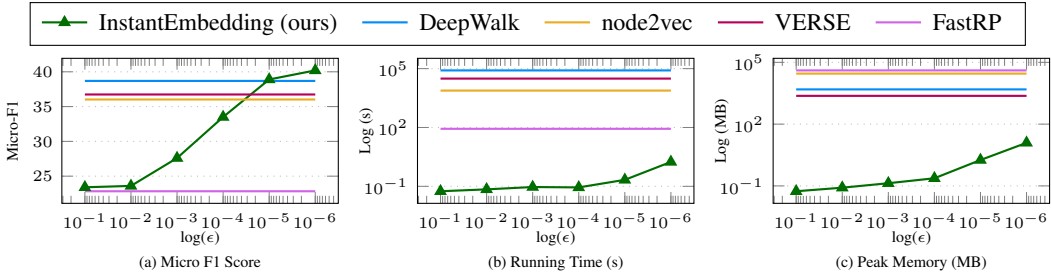

(a) Micro F1 Score      (b) Running Time (s)      (c) Peak Memory (MB)

Figure 2: The impact of the choice of $\epsilon$ on the quality of the resulting embedding (through the Micro-F1 score), average running time and peak memory increase for the YouTube dataset.

**Link prediction.** We conduct link prediction experiments to assess the capability of the produced representations to model hidden connections in the graph. For the dataset which has temporal information (CoAuthor), we select data until 2014 as training data, and split co-authorship links between 2015-2016 in two balanced partitions that we use as validation and test. For the other datasets, we uniformly sample 80% of the available edges as training (to learn embeddings on), and use the rest for validation (10%) and testing (10%). Over repeated runs, we vary the splits. More details about the experimental design are available in the supplementary material. We report results for each method in in Table 4, which shows average ROC-AUC and confidence intervals for each method. Across the datasets, our proposed method beats all baselines except VERSE, however we do achieve the best performance on YouTube by a statistically significant margin.

**Visualization.** Figure 3 presents UMAP (McInnes et al., 2018) projections on the CoCit dataset, where we grouped together similar conferences. We note that our sublinear approach is especially

well suited to visualizing graph data, as visualization algorithms only require a small subset of points (typically downsampling to only thousands) to generate a visualization for datasets.

The experimental analysis of node classification, link prediction, and visualization show that despite relying on two different approximations (PPR & random projection), InstantEmbedding is able to very quickly produce representations which meet or exceed the state of the art in unsupervised representation learning for graph structure, confirming hypothesis **H2**. We remark that interestingly InstantEmbedding seems slightly better at node classifications than link prediction. We suspect that the randomization may effectively act as a regularization which is more useful on classification.

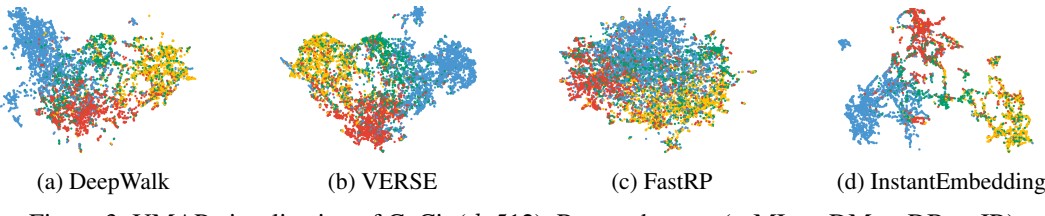

| (a) DeepWalk | (b) VERSE | (c) FastRP | (d) InstantEmbedding |

Figure 3: UMAP visualization of CoCit ($d$=512). Research areas (■ ML, ■ DM, ■ DB, ■ IR).

## 5 CONCLUSION

The present work has two main contribution: a) introducing and formally defining the *Local Node Embedding* problem and b) presenting InstantEmbedding, a highly efficient method that selectively embeds nodes using only local information, effectively solving the aforementioned problem. As existing graph embedding methods require accessing the global graph structure at least once during the representation generating process, the novelty brought by InstantEmbedding is especially impactful in real-world scenarios where graphs outgrow the capabilities of a single machine, and annotated data is scarce or expensive to produce. Embedding selectively only the critical subset of nodes for a task makes many more applications feasible in practice, while reducing the costs for others.

Furthermore, we show theoretically that our method embeds a single node in space and time sublinear to the size of the graph. We also empirically prove that InstantEmbedding is capable of surpassing state-of-the-art methods, while being many orders of magnitude faster than them – our experiments show that we are over 9,000 times faster on large datasets and on a graph over 1 billion edges we can compute a representation in 80ms.

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

# APPENDIX

## A PROOFS

### A.1 VERSE AS MATRIX FACTORIZATION

**Lemma A.1.** *(restated from Tsitsulin et al. (2020), (ref. Lemma 3.1)) VERSE implicitly factorizes the matrix* $\log(\mathbf{PPR}) + \log n - \log b$ *into* $\mathbf{XX}^\top$*, where* $n$ *is the number of nodes in the graph and* $b$ *is the number of negative samples.*

*Proof.* (from Tsitsulin et al. (2020), following Levy & Goldberg (2014); Qiu et al. (2018)) Since $\mathbf{PPR}$ is right-stochastic and the noise distribution does not depend on $j$ we can decompose the positive and negative terms from the objective of VERSE:

$$\mathcal{L} = \sum_{i=1}^n \sum_{j=1}^n \mathbf{PPR}_{ij} \log \sigma \left( \mathbf{x}_i^\top \mathbf{x}_j \right) + \frac{b}{n} \sum_{i=1}^n \sum_{j'=1}^n \log \sigma \left( -\mathbf{x}_i^\top \mathbf{x}_{j'} \right).$$

Isolating the loss for a pair of vertices $i, j$:

$$\mathcal{L}_{ij} = \mathbf{PPR}_{ij} \log \sigma \left( \mathbf{x}_i^\top \mathbf{x}_j \right) + \frac{b}{n} \log \sigma \left( -\mathbf{x}_i^\top \mathbf{x}_j \right).$$

We substitute $z_{ij} = \mathbf{x}_i^\top \mathbf{x}_j$, use our independence assumption, and solve for $\frac{\partial \mathcal{L}_{ij}}{\partial z_{ij}} = \mathbf{PPR}_{ij}\sigma(-z_{ij}) - \frac{b}{n}\sigma(z_{ij}) = 0$ to get $z_{ij} = \log \frac{n \cdot \mathbf{PPR}_{ij}}{b}$, hence $\log(\mathbf{PPR}) + \log n - \log b = \mathbf{XX}^\top$. $\qquad\square$

### A.2 HASH KERNEL

**Lemma A.2.** *(restated from Weinberger et al. (2009)) The hash kernel is unbiased:*
$$\mathbb{E}_{h_\mathrm{d} \sim \mathbb{U}_d, h_\mathrm{sgn} \sim \mathbb{U}_{-1,1}} \left[ H_{h_\mathrm{d}, h_\mathrm{sgn}}(\mathbf{x})^\top H_{h_\mathrm{d}, h_\mathrm{sgn}}(\mathbf{x}) \right] = \mathbf{x}^\top \mathbf{x}$$

**Corollary A.2.1.** *(ref. Lemma 3.2) The space complexity of* $H_{h_\mathrm{d}, h_\mathrm{sgn}}$ *is* $\mathcal{O}(1)$ *and:*
$$\mathbb{E}_{h_\mathrm{d} \sim \mathbb{U}_d, h_\mathrm{sgn} \sim \mathbb{U}_{-1,1}} \left[ H_{h_\mathrm{d}, h_\mathrm{sgn}}(\mathbf{M}) H_{h_\mathrm{d}, h_\mathrm{sgn}}(\mathbf{M})^\top \right] = \mathbf{MM}^\top$$

*Proof.* We note that the space complexity required to store a hash function from an universal family is $\mathcal{O}(1)$. Indeed, one can choose an universal hash family such that its elements are uniquely determined by a fixed choice of keys. As an example, the multiplication hash function (Cormen et al. (2009)) $h^A(x) = \lceil n(xA \bmod 1) \rceil$ requires constant memory to store the key $A \in (0, 1)$.

In order to prove the projection provides unbiased dot-products, considering the expectation per each entry, we have:

$$\mathbb{E}_{h_\mathrm{d} \sim \mathbb{U}_d, h_\mathrm{sgn} \sim \mathbb{U}_{-1,1}} \left[ \left( H_{h_\mathrm{d}, h_\mathrm{sgn}}(\mathbf{M}) H_{h_\mathrm{d}, h_\mathrm{sgn}}(\mathbf{M})^\top \right)_{i,j} \right]$$
$$= \mathbb{E}_{h_\mathrm{d} \sim \mathbb{U}_d, h_\mathrm{sgn} \sim \mathbb{U}_{-1,1}} \left[ \left( H_{h_\mathrm{d}, h_\mathrm{sgn}}(\mathbf{M}_i) H_{h_\mathrm{d}, h_\mathrm{sgn}}(\mathbf{M}_j)^\top \right) \right]$$
$$= \mathbf{M}_i \mathbf{M}_j^\top \qquad \text{From Lemma A.2}$$
$$= \left( \mathbf{MM}^\top \right)_{i,j}$$

which holds for all $i, j$ pairs. $\qquad\square$

### A.3 SPARSEPERSONALIZEDPAGERANK PROPERTIES

**Theorem A.3.** *(restated from Andersen et al. (2007)) Properties of the* SPARSEPERSONALIZEDPAGERANK$(v, G, \epsilon)$ *(3) algorithm are as follows. For any starting vector* $v$*, and any constant* $\epsilon \in (0, 1]$*, the algorithm computes an* $\epsilon$*-approximate PersonalizedPageRank vector* $p$*. Furthermore the support of* $p$ *satisfies* $vol(Supp(p)) \le \mathcal{O}\left( 1/(1-\alpha)\epsilon \right)$*, and the running time of the algorithm is* $\mathcal{O}(1/\alpha\epsilon)$*.*

We note here that Andersen et al. (2007) prove their results for the lazy transition matrix and not the standard transition matrix that we consider here. Nevertheless as discussed in Appendix A.7 switching between the two definitions does not change the asymptotic of their results.

---

**Algorithm 3** SPARSEPERSONALIZEDPAGERANK cf. Andersen et al. (2007)

---

    **Input:** node $v$, graph $G$, precision $\epsilon$, return probability $\alpha$
    **Output:** PPR vector $\pi$
1: **function** SPARSEPERSONALIZEDPAGERANK$(v, G, \epsilon, \alpha)$
2:      $\mathbf{r} \leftarrow \mathbf{0_n} \ (sparse)$
3:      $\pi \leftarrow \mathbf{0_n} \ (sparse)$
4:      $\mathbf{r}[v] = 1$
5:      **while** $\exists \, w \in G, \mathbf{r}[w] > \epsilon \times deg(w)$ **do**
6:         $\hat{r} \leftarrow \mathbf{r}[w]$
7:         $\pi[w] \leftarrow \pi[w] + \alpha\hat{r}$
8:         $\mathbf{r}[w] \leftarrow \frac{(1-\alpha)\hat{r}}{2}$
9:         $\mathbf{r}[u] \leftarrow \mathbf{r}[u] + \frac{(1-\alpha)\hat{r}}{2 \, deg(w)}, \forall (w, u) \in G$
10:     **return** $\pi$

---

## A.4 INSTANTEMBEDDING LOCALITY

**Lemma A.4.** *(ref. Lemma 3.3) The* INSTANTEMBEDDING$(v, G, d, \epsilon)$ *algorithm computes the local embedding of a node* $v$ *by exploring at most the* $\mathcal{O}\left(1/(1-\alpha)\epsilon\right)$ *nodes in the neighborhood of* $v$.

*Proof.* First recall that the only operation that explores the graph in INSTANTEMBEDDING is SPARSEPERSONALIZEDPAGERANK, which explores a node $w$ in the graph if and only if a neighbor of $w$ has a positive score (i.e. $r[w'] > \epsilon \times deg(w')$ was true and thus $\pi[w'] > 0$), and so it is part of the support of $\pi$. Furthermore at the beginning of the algorithm only $v$ is active. So it follows that every node explored by the algorithm is connected to $v$ via a path composed only by the nodes with $\pi$ score strictly larger than 0. So its distance from $v$ is bounded by the support of the $\pi$ vector that is $O\left(1/(1-\alpha)\epsilon\right)$ cf. Theorem A.3. $\square$

## A.5 INSTANTEMBEDDING COMPLEXITY

**Theorem A.5.** *(ref. Theorem 3.4) The* INSTANTEMBEDDING$(v, G, d, \epsilon)$ *algorithm has running time* $\mathcal{O}\left(d + 1/\alpha(1-\alpha)\epsilon\right)$.

*Proof.* The first step of the INSTANTEMBEDDING is computing the approximate Personalized PageRank vector. As noted in Theorem A.3, this can be done in time $\mathcal{O}\left(1/\alpha\epsilon\right)$.

We now focus our attention to the second part of our algorithm, projecting the sparse PPR vector into the embedding space. For each non-zero entry $r_j$ of the PPR vector $\pi$, we compute hash functions $h_d(j)$, $h_{sgn}(j)$ and $\max(\log(r_j * n), 0)$ in $\mathcal{O}(1)$ time. The total number of iterations is equal to the support size of $\pi$, i.e. $\mathcal{O}\left(1/(1-\alpha)\epsilon\right)$.

Finally, we note that our algorithm always generates a dense embedding, handling this variable in $O(d)$ time complexity. However, in practice this term is negligible as $1/e >> d$. Summing up the aforementioned bounds we get the total running time of our algorithm:

$$\mathcal{O}\left(d + 1/\alpha\epsilon + 1/(1-\alpha)\epsilon\right) = \mathcal{O}\left(d + 1/\alpha(1-\alpha)\epsilon\right)$$

$\square$

## A.6 GLOBAL CONSISTENCY

**Theorem A.6.** *(ref. Theorem 3.5)*
INSTANTEMBEDDING$(v, G, d, \epsilon)$ *output equals* GRAPHEMBEDDING$(G, d, \epsilon)_v$.

*Proof.* We begin by noting that for a fixed parameterization, the SPARSEPERSONALIZEDPAGERANK routine will compute an unique vector for a given node. Analyzing now the $\mathbf{W}_{v,j}$ entry of the embedding matrix generated by GRAPHEMBEDDING$(G, d, \epsilon)$, we have:

$$\mathbf{W}_{v,j} = \sum_{r_k \in \pi_v} h_{sgn}(k) \times \max(\log(r_k * n), 0)\mathbb{I}[h_d(k) = j]$$

The entire computation is deterministic and directly dependent only on the hash functions of choice and the indexing of the graph. By fixing the two hash functions $h_d$ and $h_{sgn}$, we also have that $\mathbf{W}_{v,j} = \mathbf{w}_j^v$ where $\mathbf{w}^v = \text{INSTANTEMBEDDING}(v, G, d, \epsilon), \forall v \in [0..n-1], j \in [0..d-1]$. $\quad\square$

### A.7 REPARAMETERIZATION

We note that Andersen et al. (2007) in their paper use a lazy random walk transition matrix. Furthermore in their analysis they also consider a lazy random walk. Nevertheless, this does not affect the asymptotic of their results, in fact in Proposition 8.1 in Andersen et al. (2007) they show that the two definition are equivalent up to a small change in $\alpha$. More precisely, a standard personalized PageRank with decay factor $\alpha$ is equivalent to a lazy random walk with decay factor $\frac{\alpha}{2-\alpha}$. So all the asymptotic of the bounds in Andersen et al. (2007) apply also to the classic random walk setting that we study in this paper.

## B  EXPERIMENTS

### B.1  METHODS DESCRIPTIONS

We ran all baselines on 128 and 512 embedding dimensions. As we expect our method to perform better as we increase the projection size, we performed an auxiliary test with embedding size 2048 for InstantEmbedding. We also make the observation that learning-based methods generally do not scale well with an increase of the embedding space. The following are the description and individual parameterization for each method.

- **DeepWalk** (Perozzi et al., 2014): Constructs node-contexts from random-walks and learns representations by increasing the nodes co-occurrence likelihood by modeling the posterior distribution with hierarchical softmax. We set the number of walks per node and their length to 80, and context windows size to 10.

- **Node2Vec** (Grover & Leskovec, 2016): Samples random paths in the graph similar to DeepWalk, while adding two parameters, $p$ and $q$, controlling the behaviour of the walk. Estimates the likelihood through negative sampling. We set again the number of walks per node and their length to 80 and windows size 10, number of negative samples to 5 per node and $p = q = 1$.

- **Verse** (Tsitsulin et al., 2018): Minimizes objective through gradient descent, by sampling nodes from PPR random walks and negatives samples from a noise distribution. We train it over $10^5$ epochs and set the stopping probability to 0.15.

- **FastRP** (Chen et al., 2019): Computes a high-order similarity matrix as a linear combination of multiple-steps transitions matrices and projects it into an embedding space through a sparse random matrix. We fix the linear coefficients to $[0, 0, 1, 6]$ and the normalization parameter $-0.65$.

- **InstantEmbedding** (this work): Approximate per-node PPR vectors with return probability $\alpha$ and precision $\epsilon$, which are projected into the embedding space using two fixed hash functions. In all our experiments, we set $\alpha = 0.15$ and $\epsilon > \frac{1}{n}$, where $n$ is the number of nodes in the graph.

Four additional baselines were considered for extending a subset of our experiments, as follows:

- **RandNE** (Zhang et al., 2018): Linearnly combine transition matrices of multiple orders, randomly projected through an orthogonal Gaussian matrix. We used transitions up to order 3, with coefficients $[1, 100, 10^4, 10^5]$.

- **NodeSketch** (Yang et al., 2019): Employs a recursive sketching process using a hash function that preserves hamming distances. Where provided, we use the recommended default parameters, and on all other datasets we choose the best performing ('order', $\alpha$) parameters from $\{(2, 0.0001), (5, 0.001)\}$.

- **LouvainNE** (Bhowmick et al., 2020): Aggregates node representations from a successive sub-graph hierarchy. We use the recommended defaults across all datasets, with Louvain partition strategy and a damping parameter $a = 0.01$.

Table 5: Analysis of employed networks in terms of scale-free and small-world measures. The scale-free degree is reported as a Kolmogorov-Smirnov test between power-law and exponential/log-normal distributions candidates (R = mean log-likelihood ratio, p = degree of confidence).

|  | Exponential Distribution | | Log-normal Distribution | | Pseudo-diameter | Transitivity |
|---|---|---|---|---|---|---|
|  | R | p | R | p | | |
| PPI | 12.49 | 0.02 | -0.56 | 0.45 | 8 | 0.09 |
| Blogcatalog | 63.39 | 0 | -1.9 | 0.16 | 5 | 0.09 |
| CoCit | 354.09 | 0 | -2.61 | 0.19 | 25 | 0.08 |
| CoAut | 502 | 0 | -102 | 0 | 30 | 0.28 |
| Flickr | -61.31 | 0.28 | -504 | 0 | 6 | 0.18 |
| YouTube | 60175 | 0 | -142 | 0 | 24 | 0.006 |
| Amazon2M | 41233 | 0 | -190 | 0 | 29 | 0.13 |
| Orkut | 33757 | 0 | -737 | 0 | 10 | 0.04 |

- **FREDE** (Tsitsulin et al., 2020): Sketches matrix 2 by iteratively computing per-node PPR vectors and using frequent directions. We use the provided default parameters for all datasets in order to measure running times.

## B.2 DATASET DESCRIPTIONS

The graph datasets we used in our experiments are as follows:

- *PPI* (Stark et al., 2006): Subgraph of the protein-protein interaction for Homo Sapiens species and ground-truth labels represented by biological states. Data originally processed by Grover & Leskovec (2016).

- *Blogcatalog* (Tang & Liu, 2010): Network of social interactions between bloggers. Authors specify categories for their blog, which we use as labels.

- *Microsoft Academic Graph (MAG)* (mag, 2016): Collection of scientific papers, authors, journals and conferences. Two distinct subgraphs were originally processed by Tsitsulin et al. (2018), based on co-authorship (CoAuthor) and co-citations (CoCit) relations. For the latter one, labels are represented by the unique conference where the paper was published.

- *Flickr* (Tang & Liu, 2010): Contact network of users within 195 randomly sampled interest groups.

- *YouTube* (Tang & Liu, 2010) Social network of users on the video-sharing platform. Labels are represented by group of interests with at least 500 subscribers.

- *Amazon2M* (Chiang et al., 2019): Network of products where edges are represented by co-purchased relations.

- *Orkut* (Yang & Leskovec, 2015): Social network where users can create and join groups, used at ground-truth labels. We followed the approach of Tsitsulin et al. (2018) and selected only the top 50 largest groups.

- *Friendster* (Yang & Leskovec, 2015): Social network where users can form friendship edge each other. It also allows users form a group which other members can then join.

To better understand the variety of our chosen datasets, we report the scale-free and small-world characteristics of the networks in Table 5. fitting a power-law distribution to node degrees, and comparing it to other 2 distributions (exponential and log-normal) through the Kolmogorov-Smirnov test (R = mean log-likelihood ratio, p = degree of confidence). We note that on the CoAuthorship, Flickr, YouTube, Amazon2M and Orkut networks, the h0 hypothesis can be rejected ($p < 0.05$) and thus conclude that the log-normal distribution is a better fit (graphs are not scale-free). Additionally, we report two small-world related measures: the pseudo-diameter of the graphs and their global clustering coefficient (transitivity). We observe that that graphs we use in the study are diverse, covering the spectrum of small-world and large-diameter networks.

## B.3 RUNTIME ANALYSIS

### B.3.1 GENERAL SETUP

For the runtime analysis we use the same parameterization as described in B.1 for all methods. In the special case of InstantEmbedding, we dynamically load into memory just the required subgraph in order to approximate the PPR vector for a single node. We individually ran each method on a virtual machine hosted on the Google Cloud Platform, with a 2.3GHz 16-core CPU and 128GB of RAM.

### B.3.2 RUNTIME: SPEED

All methods, except the single-threaded FastRP, leveraged the 16 cores of our machines. Some methods did not complete all the tasks: none ran on Friendster; node2vec was unable to run on Amazon2M and Orkut; FastRP did not run on Orkut specifically for a 512-dimension embedding. We note that all reported numbers are real execution times, taking into account also loading the data in memory. The detailed results are shown in Table 7.

For reference, we also provide the total running time for producing a full graph embedding. We note that when computing the PPR matrix (of an entire graph) a local measure may be suboptimal, and we leave optimizing global run time as future work. Nevertheless, here we report the total running time of our local method successively applied to all nodes in a graph in Table 6 , with an additional 4 recent baselines. All methods ran on the same machine and produced a 512-dimensional embedding for the node classification task. From the additional baselines, only RandNE and LouvainNE could scale to Orkut, while FREDE could only produce an embedding on half of the datasets.

### B.3.3 TASK: MEMORY USAGE

The methods that failed to complete in the *Running Times* section are also marked here accordingly. We note that due to the local nature of our method, we can consistently keep the average memory usage under 1MB for all datasets. This observation reinforces the sublinear guarantees of our algorithm when being within a good $\epsilon$-regime, as stated in Lemma A.4. The detailed results are shown in Table 8.

Table 6: Total approximate running time for producing a 512-dimensional full graph embedding, with 4 additional recent baselines. In this scenario, InstantEmbedding produced a full graph embedding, as opposed to the originally proposed single node representation task.

|                 | PPI    | BlogCatalog | CoCit  | Flickr | YouTube | Orkut   |
|-----------------|--------|-------------|--------|--------|---------|---------|
| DeepWalk        | 254    | 711         | 2,767  | 6,035  | 81,168  | 219,003 |
| node2vec        | 24.82  | 67.8        | 280    | 802    | 7,600   | DNC     |
| VERSE           | 87.5   | 198         | 904    | 1,863  | 31,101  | 84,468  |
| RandNE          | 2.05   | 3.93        | **3.86** | 39.4  | **32.1** | 773     |
| FastRP          | 1.81   | 5.62        | 7.21   | 79.8   | 85.5    | DNC     |
| NodeSketch      | 19.9   | 609         | 496    | 3,350  | 2,798   | DNC     |
| LouvainNE       | 0.62   | **2.47**    | 7.6    | **17.7** | 196     | **733** |
| FREDE           | 150    | 1,194       | 10,954 | DNC    | DNC     | DNC     |
| InstantEmbedding | **0.27** | 3.98      | 5.32   | 53.6   | 1,254   | 58,070  |

Table 7: Average run time (in seconds) to generate a 128-size and a 512-size node embedding for each method and each dataset with the respective standard deviation. Each experiment was run 5 times for all the methods (given their global property) except for InstantEmbedding for which we ran the experiment 1000 times (given the method's locality property).
**bold** - improvement over the baselines; *DNC* - Did Not Complete.

| | | InstantEmbedding | DeepWalk | node2vec | VERSE | FastRP |
|---|---|---|---|---|---|---|
| PPI | 128 | **0.00735** ± *0.00130* | 92.74 ± *0.68* | 12.90 ± *0.26* | 40.05 ± *0.08* | 1.42 ± *0.02* |
| | 512 | **0.00751** ± *0.00137* | 254.31 ± *7.68* | 24.82 ± *0.17* | 87.53 ± *0.24* | 1.81 ± *0.02* |
| BlogCatalog | 128 | **0.00627** ± *0.00221* | 349.66 ± *30.03* | 37.10 ± *0.19* | 109.15 ± *1.20* | 3.03 ± *0.08* |
| | 512 | **0.00826** ± *0.00436* | 711.76 ± *17.81* | 67.86 ± *0.11* | 198.75 ± *1.68* | 5.62 ± *0.15* |
| CoCit | 128 | **0.01993** ± *0.01042* | 1,015.44 ± *3.23* | 149.53 ± *1.14* | 427.06 ± *4.23* | 3.51 ± *0.12* |
| | 512 | **0.02019** ± *0.01048* | 2,766.99 ± *5.71* | 280.35 ± *0.82* | 904.53 ± *7.89* | 7.21 ± *0.72* |
| CoAuthor | 128 | **0.01612** ± *0.00733* | 1,334.55 ± *10.84* | 189.30 ± *11.78* | 468.47 ± *1.88* | 2.71 ± *0.02* |
| | 512 | **0.01630** ± *0.00761* | 3,561.27 ± *27.86* | 339.01 ± *1.04* | 1,029.88 ± *9.96* | 5.50 ± *0.08* |
| Flickr | 128 | **0.02042** ± *0.01140* | 2,519.22 ± *121.60* | 564.71 ± *5.01* | 1,038.87 ± *11.27* | 38.41 ± *0.42* |
| | 512 | **0.02051** ± *0.01128* | 6,035.50 ± *102.25* | 802.64 ± *4.95* | 1,863.41 ± *39.82* | 79.88 ± *2.00* |
| YouTube | 128 | **0.06065** ± *0.04521* | 27,249.93 ± *1,383.18* | 4,301.05 ± *21.36* | 16,618.20 ± *282.96* | 30.44 ± *1.14* |
| | 512 | **0.06128** ± *0.04534* | 81,168.81 ± *2,752.42* | 7,600.46 ± *64.14* | 31,101.92 ± *121.03* | 85.52 ± *4.81* |
| Amazon2M | 128 | **0.09746** ± *0.05306* | 63,525.32 ± *164.83* | *DNC* | 38,627.77 ± *4,058.04* | 450.84 ± *21.07* |
| | 512 | **0.09715** ± *0.05187* | 173,966.97 ± *1,094.66* | *DNC* | 73,993.64 ± *2,110.29* | 940.88 ± *31.16* |
| Orkut | 128 | **0.17192** ± *0.04782* | 94,217.21 ± *1,121.94* | *DNC* | 50,516.07 ± *4,082.24* | 843.46 ± *17.69* |
| | 512 | **0.17231** ± *0.04806* | 219,003.92 ± *781.12* | *DNC* | 84,468.50 ± *3,407.44* | *DNC* |
| Friendster | 128 | **0.07910** ± *0.04084* | *DNC* | *DNC* | *DNC* | *DNC* |
| | 512 | **0.07930** ± *0.04090* | *DNC* | *DNC* | *DNC* | *DNC* |

Table 8: Peak memory used (in MB) to generate a 128-size and 512-size node embedding for each method and each dataset. Each experiment was run once for all the methods (given their global property) except for InstantEmbedding for which we ran the experiment 1000 times (given the method's locality property) and report the mean peak memory consumption with the respective standard deviation.
**bold** - improvement over the baselines; *DNC* - Did Not Complete.

|  |  | InstantEmbedding | DeepWalk | node2vec | VERSE | FastRP |
|---|---|---|---|---|---|---|
| PPI | 128 | **0.1692** $\pm$ *0.0214* | 4.80 | 54.02 | 2.40 | 68.17 |
|  | 512 | **0.1707** $\pm$ *0.0211* | 16.75 | 65.98 | 8.39 | 197.67 |
| BlogCatalog | 128 | **0.2040** $\pm$ *0.0788* | 14.54 | 2,970.00 | 8.08 | 150.47 |
|  | 512 | **0.2140** $\pm$ *0.0871* | 46.21 | 3,000.00 | 23.92 | 504.65 |
| CoCit | 128 | **0.2697** $\pm$ *0.0848* | 52.27 | 148.93 | 24.38 | 438.27 |
|  | 512 | **0.2780** $\pm$ *0.0692* | 187.54 | 284.20 | 92.01 | 1,660.00 |
| CoAuthor | 128 | **0.1778** $\pm$ *0.0592* | 61.13 | 120.56 | 28.25 | 465.47 |
|  | 512 | **0.1803** $\pm$ *0.0642* | 220.32 | 279.75 | 107.85 | 1,770.00 |
| Flickr | 128 | **0.4138** $\pm$ *0.1525* | 140.33 | 69,860.00 | 88.83 | 1,080.00 |
|  | 512 | **0.4451** $\pm$ *0.1729* | 387.67 | 70,110.00 | 212.50 | 3,830.00 |
| YouTube | 128 | **0.5902** $\pm$ *0.2407* | 1,360.00 | 24,910.00 | 611.48 | 10,240.00 |
|  | 512 | **0.5456** $\pm$ *0.2642* | 4,860.00 | 28,410.00 | 2,360.00 | 40,610.00 |
| Amazon2M | 128 | **0.6321** $\pm$ *0.3122* | 3,380.00 | *DNC* | 1,760.00 | 26,440.00 |
|  | 512 | **0.6778** $\pm$ *0.2862* | 10,910.00 | *DNC* | 5,520.00 | 125,870.00 |
| Orkut | 128 | **0.9124** $\pm$ *0.0672* | 4,560.00 | *DNC* | 2,520.00 | 35,940.00 |
|  | 512 | **0.8884** $\pm$ *0.1224* | 14,000.00 | *DNC* | 7,240.00 | *DNC* |
| Friendster | 128 | **0.6818** $\pm$ *0.2476* | *DNC* | *DNC* | *DNC* | *DNC* |
|  | 512 | **0.7892** $\pm$ *0.1753* | *DNC* | *DNC* | *DNC* | *DNC* |

## B.4 EMBEDDING QUALITY

### B.4.1 QUALITY: NODE CLASSIFICATION

These tasks aim to measure the semantic information preserved by the embeddings, through the means of the generalization capacity of a simple classifier, trained on a small fraction of labeled representations. All methods use 512 embedding dimensions. For each methods, we perform three different splits of the data, and for our method we generate five embeddings, each time sampling a different projection matrix. We use a logistic regression (LR) classifier from using Scikit-Learn (Pedregosa et al., 2011) to train the classifiers. In the case of multi-class classification (we follow Perozzi et al. (2014) and use a one-vs-rest LR ensemble) – we assume the number of correct labels K is known and select the top K probabilities from the ensemble. To simulate the sparsity of labels in the real-wold, we train on $10\%$ of the available labels for PPI and Blogcatalog and only $1\%$ for the rest of them, while testing on the rest.

We treat CoCit as a multi-class problem as each paper is associated an unique conference were it was published. Also, for Orkut we follow the approach from Tsitsulin et al. (2018) and select only the top 50 largest communities, while further filtering nodes belonging to more than one community. In these two cases, are fitting a simply logistic regression model on the available labeled nodes. The other datasets have multiple labels per node, and we are using a One-vs-The-Rest ensemble. When evaluating, we assume the number of correct labels, $K$, is known and select the top $K$ probabilities from the ensemble. For each methods, we are performing three different splits of the data, and for our method we generate five embeddings, sampling different projection matrices.

We report the average and 90% confidence interval for micro and macro F1-scores at different fractions of known labels. The following datasets are detailed for node classification: PPI (Table 9), BlogCatalog (Table 10), CoCit (Table 11), Flickr (Table 12), and YouTube (Table 13).

We also report experiments with 4 additional baselines in Table 14. The classification task is the same, however for NodeSketch we used the recommended SVC classifier with hamming kernel, as the Logistic Regression could not infer a separation boundary in this particular case. Additionally, for FREDE we are referencing the scores from the original paper, having a comparable evaluation framework. We note that although FREDE produces better scores, it does not scale past a medium-sized graph, and its extremely high running times (Table 6) takes this approach out of the original scope of our paper.

Table 9: Classification micro and macro F1-scores for PPI.

| Method | d | Labeled Nodes | | | | | |
| | | 10.00% | | 50.00% | | 90.00% | |
| | | Micro-F1 | Macro-F1 | Micro-F1 | Macro-F1 | Micro-F1 | Macro-F1 |
|---|---|---|---|---|---|---|---|
| DeepWalk | 128 | $15.72 \pm 1.75$ | $12.56 \pm 1.84$ | $21.34 \pm 1.20$ | $18.59 \pm 1.40$ | $24.44 \pm 0.32$ | $20.36 \pm 2.74$ |
| | 512 | $16.08 \pm 0.64$ | $12.89 \pm 1.66$ | $19.90 \pm 1.02$ | $18.08 \pm 1.11$ | $21.51 \pm 5.75$ | $20.36 \pm 5.05$ |
| node2vec | 128 | $15.65 \pm 1.46$ | $12.07 \pm 1.23$ | $20.97 \pm 1.26$ | $17.86 \pm 0.85$ | $23.99 \pm 5.84$ | $19.05 \pm 2.25$ |
| | 512 | $15.03 \pm 3.18$ | $12.19 \pm 2.34$ | $21.04 \pm 1.90$ | $18.11 \pm 2.13$ | $22.02 \pm 1.14$ | $18.18 \pm 3.47$ |
| VERSE | 128 | $14.41 \pm 1.40$ | $11.56 \pm 1.37$ | $19.63 \pm 1.08$ | $16.95 \pm 1.61$ | $22.01 \pm 2.66$ | $18.71 \pm 0.61$ |
| | 512 | $12.59 \pm 2.54$ | $9.54 \pm 2.22$ | $13.62 \pm 0.88$ | $11.67 \pm 0.85$ | $16.00 \pm 0.26$ | $13.66 \pm 0.53$ |
| FastRP | 128 | $11.73 \pm 2.37$ | $7.24 \pm 1.49$ | $16.76 \pm 0.70$ | $11.03 \pm 1.05$ | $19.45 \pm 3.10$ | $11.70 \pm 2.98$ |
| | 512 | $15.74 \pm 2.19$ | $11.11 \pm 1.20$ | $21.19 \pm 2.25$ | $15.72 \pm 1.37$ | $21.52 \pm 5.31$ | $16.63 \pm 1.87$ |
| Instant Embedding | 128 | $15.88 \pm 1.36$ | $11.67 \pm 1.09$ | $20.51 \pm 0.70$ | $16.89 \pm 0.93$ | $21.82 \pm 2.47$ | $17.49 \pm 2.36$ |
| | 512 | $17.67 \pm 1.22$ | $13.04 \pm 1.06$ | $23.50 \pm 0.97$ | $19.84 \pm 1.34$ | $25.36 \pm 2.32$ | $21.21 \pm 2.92$ |
| | 2048 | $\mathbf{18.77} \pm 1.22$ | $\mathbf{13.76} \pm 1.41$ | $\mathbf{24.30} \pm 0.67$ | $\mathbf{20.44} \pm 0.85$ | $\mathbf{25.85} \pm 2.91$ | $\mathbf{22.03} \pm 3.84$ |

### B.4.2 TASK: LINK PREDICTION

For this task we create edge embeddings by combining node representations, and treat the problem as a binary classification. We observed that different strategies for aggregating embeddings could maximize the performance of different methods under evaluation, so we conducted an in-depth

Table 10: Classification micro and macro F1-scores for Blogcatalog.

| Method | $d$ | Labeled Nodes | | | | | |
| | | 10.00% | | 50.00% | | 90.00% | |
| | | Micro-F1 | Macro-F1 | Micro-F1 | Macro-F1 | Micro-F1 | Macro-F1 |
|---|---|---|---|---|---|---|---|
| DeepWalk | 128 | $36.05 \pm 0.85$ | $20.91 \pm 0.79$ | $41.07 \pm 1.05$ | $26.85 \pm 0.96$ | $42.69 \pm 1.49$ | $28.87 \pm 4.61$ |
| | 512 | $32.48 \pm 0.35$ | $18.69 \pm 1.17$ | $37.88 \pm 0.61$ | $25.38 \pm 0.85$ | $40.14 \pm 4.03$ | $26.11 \pm 6.42$ |
| node2vec | 128 | $33.63 \pm 0.96$ | $15.28 \pm 0.99$ | $37.18 \pm 0.82$ | $20.02 \pm 0.44$ | $38.34 \pm 3.62$ | $21.26 \pm 1.37$ |
| | 512 | $33.67 \pm 0.93$ | $16.24 \pm 1.11$ | $37.42 \pm 1.40$ | $21.43 \pm 0.73$ | $38.98 \pm 4.70$ | $21.94 \pm 1.49$ |
| VERSE | 128 | $32.57 \pm 0.96$ | $18.67 \pm 1.46$ | $38.66 \pm 0.88$ | $25.0 \pm 1.37$ | $39.47 \pm 1.34$ | $26.64 \pm 1.08$ |
| | 512 | $24.64 \pm 0.85$ | $12.33 \pm 1.58$ | $29.27 \pm 0.41$ | $18.48 \pm 0.88$ | $33.18 \pm 2.51$ | $21.11 \pm 2.60$ |
| FastRP | 128 | $28.68 \pm 0.35$ | $12.74 \pm 1.23$ | $31.22 \pm 1.34$ | $14.78 \pm 0.53$ | $31.61 \pm 1.90$ | $15.34 \pm 3.27$ |
| | 512 | $33.54 \pm 0.96$ | $17.83 \pm 1.90$ | $36.94 \pm 1.08$ | $21.49 \pm 0.38$ | $37.62 \pm 2.66$ | $22.26 \pm 2.98$ |
| Instant Embedding | 128 | $27.99 \pm 1.20$ | $13.72 \pm 1.49$ | $32.40 \pm 1.23$ | $18.77 \pm 1.40$ | $33.40 \pm 2.95$ | $19.94 \pm 3.30$ |
| | 512 | $33.36 \pm 1.11$ | $17.37 \pm 1.61$ | $37.76 \pm 1.37$ | $23.79 \pm 1.61$ | $39.33 \pm 3.45$ | $26.14 \pm 3.07$ |
| | 2048 | $\mathbf{36.05} \pm 1.66$ | $19.01 \pm 1.93$ | $\mathbf{41.42} \pm 1.49$ | $\mathbf{27.16} \pm 1.96$ | $42.46 \pm 4.35$ | $\mathbf{29.00} \pm 3.94$ |

Table 11: Classification micro and macro F1-scores for CoCit.

| Method | $d$ | Labeled Nodes | | | | | |
| | | 1.00% | | 5.00% | | 9.00% | |
| | | Micro-F1 | Macro-F1 | Micro-F1 | Macro-F1 | Micro-F1 | Macro-F1 |
|---|---|---|---|---|---|---|---|
| DeepWalk | 128 | $36.51 \pm 0.85$ | $27.54 \pm 1.26$ | $41.52 \pm 0.03$ | $29.85 \pm 1.31$ | $43.21 \pm 0.61$ | $30.31 \pm 0.50$ |
| | 512 | $37.44 \pm 0.67$ | $26.57 \pm 0.76$ | $39.41 \pm 1.11$ | $29.92 \pm 0.79$ | $40.95 \pm 0.82$ | $31.48 \pm 0.91$ |
| node2vec | 128 | $37.55 \pm 0.99$ | $26.38 \pm 0.88$ | $42.92 \pm 0.55$ | $31.12 \pm 0.41$ | $43.94 \pm 0.61$ | $32.03 \pm 0.20$ |
| | 512 | $38.35 \pm 1.75$ | $27.71 \pm 1.17$ | $42.53 \pm 0.26$ | $31.05 \pm 0.50$ | $43.99 \pm 0.32$ | $32.14 \pm 0.38$ |
| VERSE | 128 | $38.52 \pm 0.47$ | $28.17 \pm 1.20$ | $41.68 \pm 0.96$ | $31.14 \pm 0.26$ | $43.47 \pm 0.26$ | $32.22 \pm 0.53$ |
| | 512 | $38.22 \pm 1.34$ | $27.42 \pm 0.91$ | $38.03 \pm 0.58$ | $29.50 \pm 0.88$ | $38.88 \pm 0.61$ | $31.04 \pm 0.82$ |
| FastRP | 128 | $15.97 \pm 0.55$ | $4.18 \pm 0.29$ | $16.74 \pm 0.64$ | $4.31 \pm 0.47$ | $16.62 \pm 0.35$ | $4.17 \pm 0.29$ |
| | 512 | $18.88 \pm 1.28$ | $6.63 \pm 0.47$ | $26.82 \pm 1.23$ | $9.17 \pm 0.26$ | $27.91 \pm 0.99$ | $8.79 \pm 0.38$ |
| Instant Embedding | 128 | $38.19 \pm 1.07$ | $25.29 \pm 1.14$ | $41.23 \pm 0.49$ | $27.92 \pm 0.63$ | $42.48 \pm 0.42$ | $28.44 \pm 0.72$ |
| | 512 | $39.95 \pm 0.67$ | $27.64 \pm 1.22$ | $43.01 \pm 0.51$ | $30.61 \pm 0.51$ | $44.05 \pm 0.35$ | $31.50 \pm 0.63$ |
| | 2048 | $\mathbf{40.49} \pm 1.06$ | $\mathbf{28.86} \pm 0.81$ | $\mathbf{43.79} \pm 0.46$ | $\mathbf{31.69} \pm 0.55$ | $\mathbf{44.85} \pm 0.46$ | $\mathbf{32.76} \pm 0.41$ |

Table 12: Classification micro and macro F1-scores for Flickr.

| Method | $d$ | Labeled Nodes | | | | | |
| | | 1.00% | | 5.00% | | 9.00% | |
| | | Micro-F1 | Macro-F1 | Micro-F1 | Macro-F1 | Micro-F1 | Macro-F1 |
|---|---|---|---|---|---|---|---|
| DeepWalk | 128 | $32.55 \pm 0.91$ | $13.81 \pm 1.72$ | $37.44 \pm 0.44$ | $22.58 \pm 0.53$ | $38.78 \pm 0.23$ | $24.75 \pm 0.58$ |
| | 512 | $31.22 \pm 0.38$ | $13.42 \pm 1.23$ | $35.67 \pm 0.38$ | $22.72 \pm 1.52$ | $37.25 \pm 0.09$ | $25.74 \pm 0.58$ |
| node2vec | 128 | $29.27 \pm 0.96$ | $6.40 \pm 0.50$ | $34.12 \pm 0.47$ | $12.82 \pm 0.88$ | $35.15 \pm 0.03$ | $14.89 \pm 0.47$ |
| | 512 | $29.80 \pm 0.67$ | $7.14 \pm 0.20$ | $34.40 \pm 0.26$ | $13.50 \pm 0.20$ | $35.39 \pm 0.06$ | $15.58 \pm 0.58$ |
| VERSE | 128 | $28.04 \pm 1.84$ | $10.52 \pm 2.37$ | $33.52 \pm 0.12$ | $19.12 \pm 0.41$ | $35.38 \pm 0.41$ | $22.31 \pm 0.93$ |
| | 512 | $25.22 \pm 0.20$ | $7.20 \pm 1.28$ | $28.25 \pm 0.29$ | $14.17 \pm 1.02$ | $29.65 \pm 0.32$ | $17.09 \pm 0.29$ |
| FastRP | 128 | $28.20 \pm 0.53$ | $9.39 \pm 1.61$ | $30.43 \pm 0.15$ | $13.82 \pm 0.61$ | $30.65 \pm 0.29$ | $14.51 \pm 0.38$ |
| | 512 | $29.85 \pm 0.26$ | $12.28 \pm 2.72$ | $33.64 \pm 0.58$ | $18.94 \pm 1.28$ | $34.88 \pm 0.58$ | $21.44 \pm 1.23$ |
| Instant Embedding | 128 | $27.41 \pm 0.90$ | $9.14 \pm 0.56$ | $31.84 \pm 0.25$ | $14.90 \pm 0.55$ | $33.14 \pm 0.33$ | $17.27 \pm 0.65$ |
| | 512 | $30.43 \pm 0.79$ | $10.78 \pm 1.20$ | $34.00 \pm 0.25$ | $18.36 \pm 0.51$ | $35.37 \pm 0.25$ | $21.26 \pm 0.48$ |
| | 2048 | $31.89 \pm 0.62$ | $11.15 \pm 1.02$ | $35.94 \pm 0.23$ | $19.38 \pm 0.85$ | $37.21 \pm 0.18$ | $23.02 \pm 0.56$ |

Table 13: Classification micro and macro F1-scores for YouTube.

| Method | $d$ | Labeled Nodes | | | | | |
| | | 1.00% | | 5.00% | | 9.00% | |
| | | Micro-F1 | Macro-F1 | Micro-F1 | Macro-F1 | Micro-F1 | Macro-F1 |
|---|---|---|---|---|---|---|---|
| DeepWalk | 128 | $37.53 \pm 1.40$ | $29.04 \pm 3.77$ | $41.64 \pm 0.15$ | $34.45 \pm 0.70$ | $42.97 \pm 0.29$ | $35.62 \pm 0.93$ |
| | 512 | $38.69 \pm 1.17$ | $31.11 \pm 1.08$ | $40.26 \pm 0.38$ | $35.09 \pm 0.26$ | $40.74 \pm 0.06$ | $36.14 \pm 0.23$ |
| VERSE | 128 | $37.13 \pm 0.41$ | $28.54 \pm 2.39$ | $39.74 \pm 0.32$ | $33.87 \pm 0.67$ | $41.70 \pm 0.38$ | $35.04 \pm 0.41$ |
| | 512 | $36.74 \pm 1.05$ | $27.16 \pm 0.15$ | $37.47 \pm 1.37$ | $32.40 \pm 0.91$ | $37.64 \pm 0.67$ | $33.00 \pm 0.35$ |
| node2vec | 128 | $34.64 \pm 2.63$ | $25.35 \pm 3.83$ | $40.62 \pm 1.02$ | $33.26 \pm 0.20$ | $42.65 \pm 0.70$ | $35.73 \pm 0.32$ |
| | 512 | $36.02 \pm 2.01$ | $25.03 \pm 2.89$ | $39.64 \pm 0.44$ | $33.78 \pm 0.38$ | $40.47 \pm 0.85$ | $35.01 \pm 1.08$ |
| FastRP | 128 | $23.61 \pm 1.61$ | $6.24 \pm 0.61$ | $24.16 \pm 0.96$ | $6.64 \pm 1.64$ | $24.50 \pm 0.29$ | $7.09 \pm 0.35$ |
| | 512 | $22.83 \pm 0.41$ | $7.21 \pm 0.20$ | $23.43 \pm 0.55$ | $8.77 \pm 0.82$ | $23.76 \pm 0.64$ | $9.56 \pm 0.91$ |
| Instant Embedding | 128 | $37.89 \pm 1.02$ | $26.27 \pm 1.36$ | $40.90 \pm 0.53$ | $31.57 \pm 0.86$ | $41.78 \pm 0.37$ | $32.73 \pm 0.51$ |
| | 512 | $40.04 \pm 0.97$ | $27.52 \pm 1.60$ | $43.31 \pm 0.41$ | $33.98 \pm 0.81$ | $44.00 \pm 0.42$ | $35.56 \pm 0.69$ |
| | 2048 | $\mathbf{40.91} \pm 0.86$ | $28.34 \pm 1.43$ | $\mathbf{44.82} \pm 0.49$ | $\mathbf{35.16} \pm 1.02$ | $\mathbf{45.67} \pm 0.32$ | $\mathbf{36.90} \pm 0.69$ |

investigation in order for the fairest possible evaluation. Specifically, for two node embeddings $w$ and $\hat{w}$ we adopt the following strategies for creating edge representations:

1. dot-product: $\mathbf{w}^\top \hat{\mathbf{w}}$
2. cosine distance: $\frac{\mathbf{w}^\top \hat{\mathbf{w}}}{\|\mathbf{w}\| \|\hat{\mathbf{w}}\|}$
3. hadamard product: $\mathbf{w} \odot \hat{\mathbf{w}}$
4. element-wise average: $\frac{1}{2}(\mathbf{w} + \hat{\mathbf{w}})$
5. L1 element-wise distance: $|\mathbf{w} - \hat{\mathbf{w}}|$
6. L2 element-wise distance $(\mathbf{w} - \hat{\mathbf{w}}) \odot (\mathbf{w} - \hat{\mathbf{w}})$

While the first two strategies directly create a ranking from two embeddings, for the other ones we train a logistic regression on examples from the validation set. In all cases, a likelihood scalar value will be attributed to all edges, and we report their ROC-AUC score on the test set.

Taking into account that different embedding methods may determine a specific topology of the embedding space, that may in turn favour a specific edge aggregation method, for each method we consider only the strategy that consistently provides good results on all datasets. This ensures that all methods can be objectively compared to one another, independent of the particularities of induced embedding space geometry.

The following tables show detailed analysis of link prediction results for BlogCatalog (Table 15) and CoAuthor (Table 16).

Table 14: Approximate Micro-F1 scores with an additional 4 baselines. All methods produced 512-dimensional embeddings, with the exception of FREDE for which we refer the scores from the original paper.

| | PPI | BlogCatalog | CoCit | Flickr | YouTube | Orkut |
|---|---|---|---|---|---|---|
| DeepWalk | 16.08 | 32.48 | 37.44 | **31.22** | 38.69 | **87.67** |
| node2vec | 15.03 | 33.67 | 38.35 | 29.80 | 36.02 | DNC |
| VERSE | 12.59 | 24.64 | 38.22 | 25.22 | 36.74 | 81.52 |
| RandNE | 15.77 | 32.79 | 16.23 | 29.19 | 24.57 | 59.01 |
| FastRP | 15.74 | 33.54 | 26.03 | 29.85 | 22.83 | DNC |
| NodeSketch | 12.03 | 29.65 | 16.67 | 23.07 | 32.02 | DNC |
| LouvainNE | 14.73 | 22.28 | 35.14 | 26.37 | 33.52 | 45.08 |
| FREDE | **19.56** | **35.69** | **42.46** | DNC | DNC | DNC |
| InstantEmbedding | 17.67 | 33.36 | 39.95 | 30.43 | **40.04** | 76.83 |

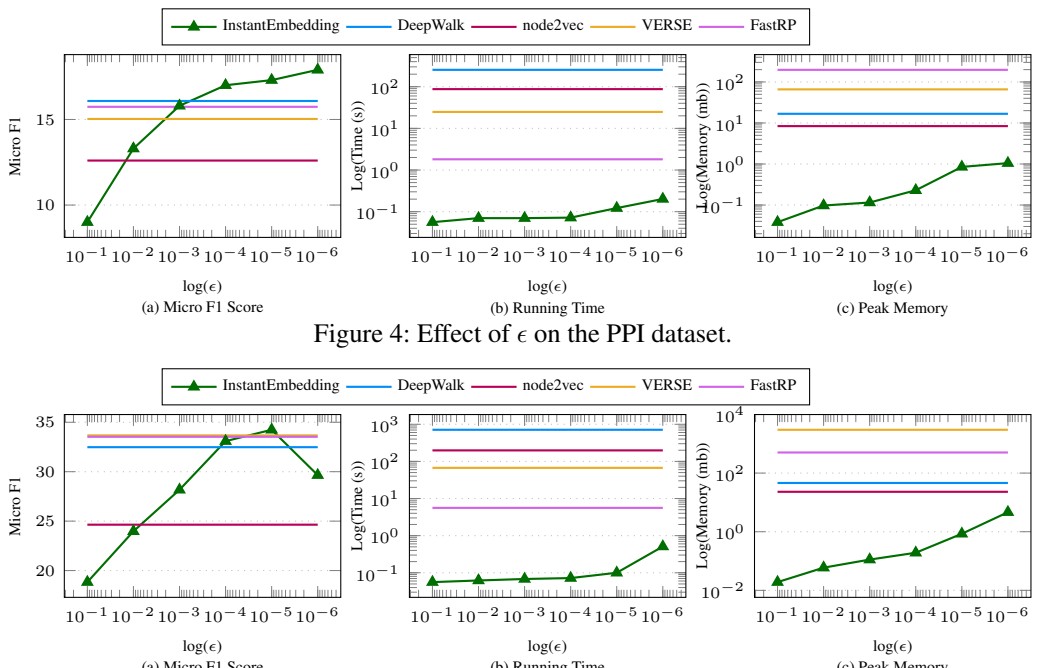

Figure 4: Effect of $\epsilon$ on the PPI dataset.

Figure 5: Effect of $\epsilon$ on the BlogCatalog dataset.

## B.5 EPSILON INFLUENCE

In order to gain insight into the effect of $\epsilon$ on the behaviour of our method, we test 6 values in the range of $[10^{-1}, ..., 10^{-6}]$. We note that the decrease of $\epsilon$ is strongly correlated with a better classification performance, but also to a larger computational overhead. The only apparent exception seems to be the Micro-F1 score on the Blogcatalog dataset, which drops suddenly when $\epsilon = 10^{-6}$. We argue that this is due to the fact that more probability mass is dispersed further away from the central node, but the max operator cuts that information away (as the number of nodes is small), and thus the resulting embedding is actually less accurate.

### B.5.1 TASK: VISUALIZATION

Figure 3 presents multiple UMAP (McInnes et al., 2018) projections on the CoCit dataset, where we grouped together similar conferences. We note that our sublinear approach is especially well suited to

Table 15: Link-prediction ROC-AUC scores for Blogcatalog. For each method, we highlight the aggregation function that consistently performs good on all datasets.

| Method | $d$ | Aggregation Function | | | | | |
| | | hadamard | dot-product | cosine | L1 | L2 | average |
|---|---|---|---|---|---|---|---|
| DeepWalk | 128 | $68.92 \pm 2.45$ | $63.01 \pm 2.83$ | $75.73 \pm 1.49$ | $91.51 \pm 0.61$ | $91.84 \pm 0.88$ | $82.07 \pm 0.09$ |
| | 512 | $67.70 \pm 1.58$ | $62.80 \pm 2.07$ | $72.83 \pm 0.82$ | $90.94 \pm 0.29$ | $91.41 \pm 0.67$ | $83.71 \pm 1.46$ |
| node2vec | 128 | $93.12 \pm 0.20$ | $91.85 \pm 1.37$ | $22.52 \pm 0.41$ | $89.90 \pm 0.70$ | $90.28 \pm 1.28$ | $94.41 \pm 0.53$ |
| | 512 | $92.18 \pm 0.12$ | $90.96 \pm 0.12$ | $12.49 \pm 1.20$ | $93.89 \pm 0.38$ | $93.50 \pm 0.76$ | $93.72 \pm 0.26$ |
| VERSE | 128 | $94.96 \pm 0.38$ | $95.10 \pm 0.67$ | $85.21 \pm 0.88$ | $75.74 \pm 0.85$ | $75.92 \pm 0.73$ | $94.07 \pm 0.47$ |
| | 512 | $93.42 \pm 0.35$ | $93.40 \pm 0.67$ | $61.48 \pm 0.88$ | $91.52 \pm 0.26$ | $92.17 \pm 0.61$ | $93.14 \pm 0.58$ |
| FastRP | 128 | $73.54 \pm 0.23$ | $68.16 \pm 0.55$ | $76.32 \pm 1.90$ | $85.78 \pm 2.31$ | $82.46 \pm 2.01$ | $89.25 \pm 0.85$ |
| | 512 | $78.34 \pm 2.80$ | $70.67 \pm 0.79$ | $79.25 \pm 1.02$ | $88.68 \pm 0.70$ | $84.56 \pm 0.76$ | $90.99 \pm 0.55$ |
| Instant Embedding | 128 | $89.22 \pm 1.48$ | $84.95 \pm 4.19$ | $51.57 \pm 1.14$ | $72.52 \pm 1.71$ | $64.39 \pm 1.37$ | $87.65 \pm 0.70$ |
| | 512 | $92.74 \pm 0.60$ | $90.77 \pm 1.51$ | $51.75 \pm 1.16$ | $83.07 \pm 1.00$ | $70.39 \pm 1.11$ | $90.63 \pm 0.56$ |
| | 2048 | $93.84 \pm 0.33$ | $93.44 \pm 0.53$ | $51.35 \pm 1.18$ | $88.95 \pm 0.85$ | $77.39 \pm 1.02$ | $92.40 \pm 0.42$ |

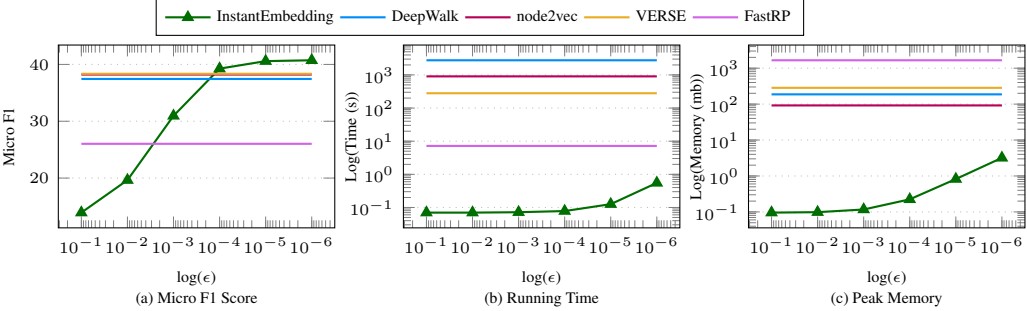

Figure 6: Effect of $\epsilon$ on the CoCit dataset.

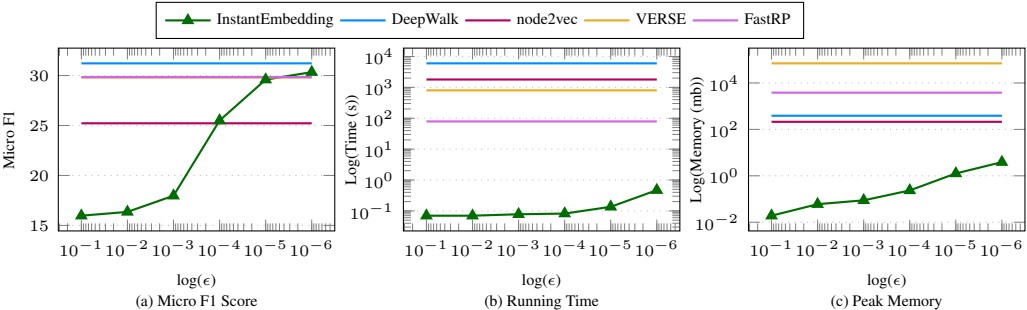

Figure 7: Effect of $\epsilon$ on the Flickr dataset.

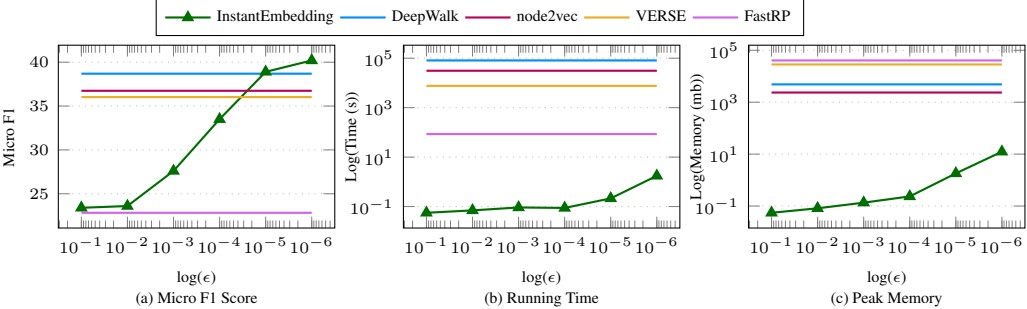

Figure 8: Effect of $\epsilon$ on the YouTube dataset.

Table 16: Temporal link-prediction ROC-AUC scores for CoAuthor. For each method, we highlight the aggregation function that consistently performs good on all datasets.

| | | Aggregation Function | | | | | |
|---|---|---|---|---|---|---|---|
| Method | d | hadamard | dot-product | cosine | L1 | L2 | average |
| DeepWalk | 128 | $75.59 \pm 0.88$ | $74.05 \pm 1.58$ | $83.5 \pm 0.12$ | $86.99 \pm 0.09$ | $87.21 \pm 0.73$ | $73.64 \pm 1.72$ |
| | 512 | $78.42 \pm 0.53$ | $76.40 \pm 1.87$ | $82.05 \pm 1.20$ | $87.85 \pm 0.29$ | $88.43 \pm 1.08$ | $79.56 \pm 0.70$ |
| node2vec | 128 | $80.18 \pm 0.67$ | $45.00 \pm 1.34$ | $54.59 \pm 0.88$ | $70.14 \pm 1.31$ | $70.32 \pm 0.58$ | $79.07 \pm 0.53$ |
| | 512 | $86.09 \pm 0.85$ | $45.19 \pm 0.20$ | $42.99 \pm 1.66$ | $72.41 \pm 1.84$ | $72.70 \pm 1.43$ | $84.00 \pm 0.38$ |
| VERSE | 128 | $93.16 \pm 0.44$ | $92.74 \pm 0.15$ | $90.85 \pm 0.20$ | $79.24 \pm 1.49$ | $80.27 \pm 0.41$ | $86.50 \pm 0.47$ |
| | 512 | $92.75 \pm 0.73$ | $92.36 \pm 1.08$ | $90.33 \pm 0.20$ | $72.58 \pm 1.17$ | $73.82 \pm 1.49$ | $86.69 \pm 1.02$ |
| FastRP | 128 | $60.23 \pm 1.78$ | $59.97 \pm 1.61$ | $65.08 \pm 0.93$ | $78.51 \pm 0.64$ | $77.66 \pm 0.23$ | $57.69 \pm 1.90$ |
| | 512 | $61.16 \pm 1.75$ | $61.92 \pm 0.85$ | $70.12 \pm 0.38$ | $82.19 \pm 2.22$ | $78.51 \pm 1.99$ | $63.87 \pm 1.49$ |
| Instant Embedding | 128 | $89.41 \pm 0.67$ | $88.88 \pm 0.79$ | $89.15 \pm 0.63$ | $66.19 \pm 1.92$ | $66.78 \pm 1.90$ | $83.22 \pm 0.86$ |
| | 512 | $90.44 \pm 0.48$ | $90.10 \pm 0.69$ | $90.60 \pm 0.55$ | $76.50 \pm 1.44$ | $75.76 \pm 1.41$ | $85.64 \pm 0.67$ |
| | 2048 | $89.45 \pm 0.62$ | $90.38 \pm 0.60$ | $90.84 \pm 0.44$ | $88.42 \pm 0.48$ | $84.83 \pm 0.67$ | $87.67 \pm 1.07$ |

visualizing graph data, as visualization algorithms (such as t-SNE or UMAP) only require a small subset of points (typically downsampling to only thousands) to generate a visualization for datasets.

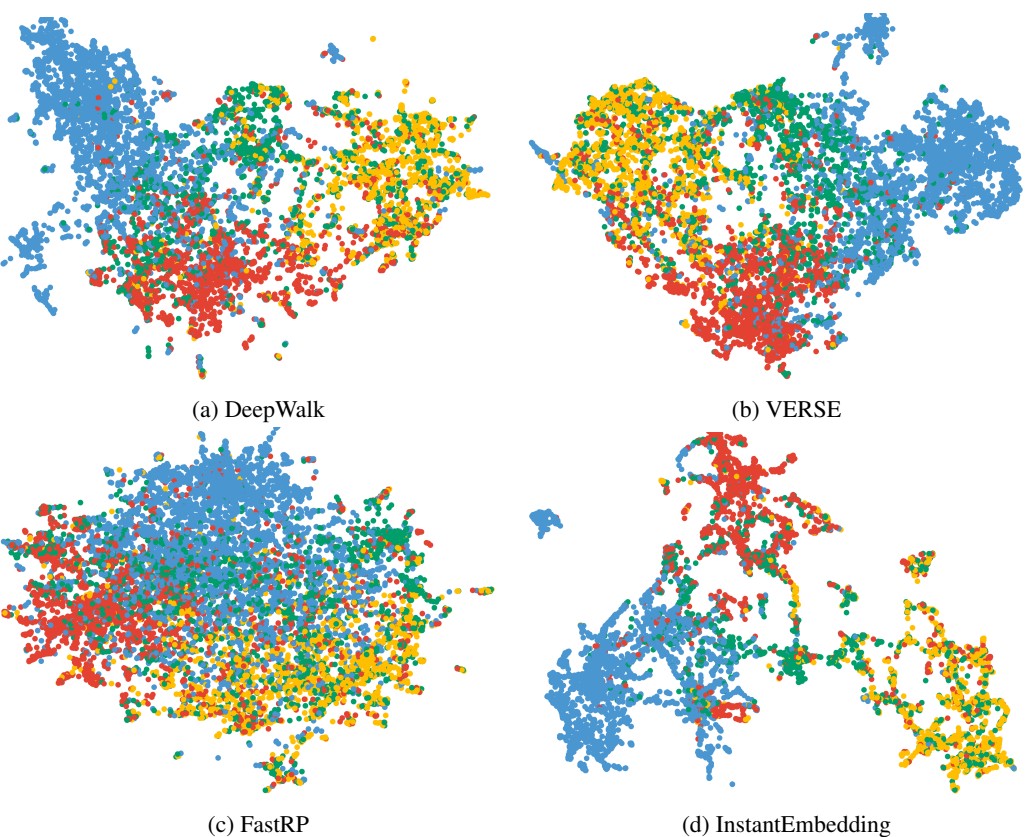

(a) DeepWalk

(b) VERSE

(c) FastRP

(d) InstantEmbedding

Figure 9: UMAP visualization of CoCit ($d$=512). Research areas (■ ML, ■ DM, ■ DB, ■ IR).

