# OpenReview forum: "InstantEmbedding: Efficient Local Node Representations"
_ICLR.cc/2021/Conference — Reject_

### Official Review · AnonReviewer3 · 2020-10-22
**Need fair comparisons**

**Rating:** 4
**Confidence:** 3

**Review:**

This paper proposes an efficient algorithm to obtain a given node's embedding based on its local PageRank scores. The proposed approach uses a hashing technique to hold similarities between nodes effectively. Besides, it exploits a local partition approach to compute PageRank scores. This paper performs the experiments by using several real datasets to show the approach’s efficiency and effectiveness.

In my opinion, the proposed approach lacks the originality; it is a combination of the existing hashing technique by Weinberger et al., and the PageRank computation approach by Andersen et al. Unlike the previous approaches such as DeepWalk, it exploits PageRank score to obtain similarities. However, this is the same idea proposed by Tsitsulin et al.

I doubt the paper performed a fair comparison to the existing approaches in terms of running time. As shown in Algorithm 2, the proposed approach computes an embedding of a single node. On the other hand, the existing embedding approaches compute embeddings of all the nodes. Therefore, the proposed approach should be evaluated in computing embeddings of all the nodes. I do not think it takes only 80ms to compute embeddings of all the nodes for Friendster dataset of 66M nodes. The downstream applications, such as visualization, need embeddings of all the nodes, not a single node. Could you tell me whether the proposed approach is faster than the previous approaches if it computes all the nodes' embeddings?

In terms of running time, it is good to compare the proposed approach to the state-of-the-art approaches such as FastRP.

Since the proposed approach removes non-significant PPR scores by using log b, it would impact the proposed approach’s running time and accuracy. Could you tell me the effects of log b?

---

> ### Author Response · Authors · 2020-11-20
> **Response to R3**
>
> We thank the reviewer for the questions and comments.
>
> First, we would like to highlight why the proposed problem makes sense to solve. Many different downstream applications only require embeddings of a handful of nodes, especially when the graphs grow larger than a couple of thousands of nodes. For node classification, we almost never have all the nodes in the graph labelled for learning - this is the case even for academic datasets used in this work - for both YouTube and Orkut only 3% of the graph is labelled. For link prediction in large graphs, the overwhelming majority of new links happen within the distance 2 or 3 to the node. Therefore, we only need to rank a small number of nodes to get the accurate link prediction estimations. For visualisation, there is almost never a need to draw millions of points to find a pattern in them; datasets are typically subsampled for visualisation purposes. For instance, in our Figure 3 we only use 20% of all nodes to showcase the visualisation capabilities, and the same patterns were obtainable with ~5% of the nodes.
>
> As requested, we report the running time of embedding all the nodes of the graph in the table below. All baselines produced 512-dimensional embeddings, using the provided default parameters. For our method, we report the total running time that provided a graph embedding of quality within 5% of the best achievable score. We remind that after a certain $\epsilon$ value, the embedding quality gains flattens while the running time increases exponentially. Though our implementation is not constantly the fastest on the full graph, we remark that its total running speed would decrease linearly with the percentage of actual nodes that need to be embedded.
>
> **Full running times**
>
> | Method/Dataset   | PPI    | BlogCatalog | CoCit | Flickr | YouTube | Orkut   |
> | :--------------- | :----- | :---------- | :---- | :----- | :------ | :------ |
> | DeepWalk         | 254    | 711         | 2,767 | 6,035  | 81,168  | 219,003 |
> | node2vec         | 24\.82 | 67\.8       | 280   | 802    | 7,600   | OOM     |
> | VERSE            | 87\.5  | 198         | 904   | 1,863  | 31,101  | 84,468  |
> | FastRP           | 1\.81  | 5\.62       | 7\.21 | 79\.8  | 85\.5   | OOM     |
> | InstantEmbedding | 0\.27  | 3\.98       | 5\.32 | 53\.6  | 1,254   | 58,070  |
>
> For the PPR filtering by log(N), for large graphs, it is smaller than our PPR approximation parameter $\epsilon$, therefore, it does not impact the embedding generation process. We keep it in order to be consistent with the PPR embedding literature, in terms of the analysis in Tsitsulin et al. (2020).

---

### Official Review · AnonReviewer2 · 2020-10-27
**Node embedding in sublinear time**

**Rating:** 6
**Confidence:** 4

**Review:**

Summary:

Contributions of this paper are twofold. First, it introduces the problem of Local Node Embedding, which means node embedding by using local information only while garanteeing global consistency. Second, it presents InstantEmbedding, an efficient implementation based on PPR and random projections. Extensive experiments on 10 datasets show that this new algorithm saves computer memory and is much faster than existing baselines.


##########################################################################

Reasons for score:

I vote for accepting the paper for I think we need such efficient approaches that are able to deal with huge graph (for industrial applications), or juste help when one uses a simple computer with limited capacities. I don't give a higher grade for I think it's a rather incremental proposal that is build on a pipeline combining two previous papers.


##########################################################################

Pros:

1. very well written paper
2. approach well grounded on two previous interesting works
3. new approach is proven to be more efficient, in term of runtime and memory usage


##########################################################################

Cons:

1. quite incremental, seems more like an application of existing methods
2. way of comparing to previous work is not perfectly clear (see below)
3. no discussion on the better performances in node classification (see below)

##########################################################################

Questions during rebuttal period:

- The way runtime is calculated in Section 4.2 is not perfectly clear. Is it calculated on the whole graph, and then divided by n (number of nodes), or is it directly at the node level? If this is the latter, how do you calculate the values for the competitors? For DeepWalk, in particular, complexity is proportional to the parameters defining the random walk, not n as shown in Table 1.

- Can you explain why DeepWalk cannot be described as a local method since it is based on truncated random walks?

- Do you have any explanation for the very good results in node classification? There is no clear reason why the proposed approach will be good for solving this task. The quick mention of regularization is not fully satisfying and further insight would be much welcome.

#########################################################################

Some typos:

"for its adaption"
"should posses"
"can choose and universal"

---

> ### Author Response · Authors · 2020-11-20
> **Response to R2**
>
> We thank the reviewer for the detailed questions and comments.
>
> Overall, we indeed propose a simple approach to the problem; it does, however, beat sophisticated state-of-the-art techniques. We emphasise that our approach is not incremental with respect to other works in the graph embedding community, as we modify the very constraints of the problem.
>
> We now answer the questions raised in the review.
>
> [1] For our method, we directly perform measurements at node level. The values for the competitors are calculated as the value for the full embedding, as this is the “minimum” computation required to generate a (globally consistent) node embedding. We do that because for the larger datasets, the labelled nodes typically constitute a small part of the whole graph. For instance, in both YouTube and Orkut the labelled nodes represent merely 3% of the whole graph. While indeed the choice of parameters does impact the running time of the competitors, we first select the best (in terms of embedding quality) configuration for a particular dataset, and then perform our measurements on multiple runs with that configuration.
>
> [2] For DeepWalk, we do not list the number of random walks per node, their length, or the window size, as these are all parameters that are set independently of the size of the graph. More importantly, the algorithm generates these walks *for each node*, resulting in O(n) complexity. In order to produce relatable node-level embeddings, Deepwalk depends on shared global state, which means one can not possibly embed a single node in isolation.
>
> [3] We believe that our method succeeds on node classification because it does a form of denoising because of the PPR truncation. This way, the embedding only consists of the strong part of the signal. Also, as seen with the visualisation task, our method provides a better separability of the embedding space, and thus the logistic regression model we imply for the classification task could generalize more easily, with fewer data.

---

> > ### Comment · AnonReviewer2 · 2020-11-24
> > **response to the response**
> >
> > [1] I'm ok but you should add the runtime you gave when responding to rev.3 in appendices. This way you give the full information on the total runtime of your approach and perform a fairer comparison.
> >
> > [2] Ok I agree you're method is purely local.
> >
> > [3] PPR truncation has already been proposed in Global Vectors for Node Representations" (https://arxiv.org/pdf/1902.11004.pdf). I think you should cite this previous work.

---

> > > ### Author Response · Authors · 2020-11-25
> > > **Response to R2**
> > >
> > > [1] Additional runtime measurements have been added in the revised version.
> > >
> > > [3] While our method is strongly related to previous work based on explicit matrix factorisation such as Qiu et al. (2018) and Brochier et al. (2019), we remark that our local PPR formulation is different from the truncated random walks by not being constrained on a predefined walk length, but rather on a maximum volume of a neighborhood.  The resulting matrix is still sparse, but the expected number of entries per row is expected to be more balanced, regardless of the density of the graph in each neighborhood. In a future work we are interested in exploring more this difference and its implications.
> > >
> > > References
> > >
> > > Qiu, J., Dong, Y., Ma, H., Li, J., Wang, K., & Tang, J. (2018, February). Network embedding as matrix factorization: Unifying deepwalk, line, pte, and node2vec. In Proceedings of the Eleventh ACM International Conference on Web Search and Data Mining (pp. 459-467).
> > >
> > > Brochier, R., Guille, A., & Velcin, J. (2019, May). Global vectors for node representations. In The World Wide Web Conference (pp. 2587-2593).

---

### Official Review · AnonReviewer4 · 2020-10-28
**Interesting scalable model for computing node embeddings. The experimental evaluation though should be enhanced.**

**Rating:** 4
**Confidence:** 5

**Review:**

###### Summary ######

The paper proposes a node representation learning method for undirected networks based on the Personalized PageRank algorithm. The embedding vector for a node is obtained by applying a hashing algorithm on the approximated PageRank vector computed by using only the local region of the interested node. That way, the paper proposes a scalable algorithm that does not require the whole network while computing the embedding of a particular node. The performance of the algorithm is evaluated on the multi-label node classification and link prediction tasks.

###### Reasons for score ######

I vote for rejection. The structure and the main idea of the paper are good, but the experimental evaluation of the method is not sufficient and it should be enhanced. Although there are many recent scalable algorithms (as described below), the paper considers only FastRP and the classical models as baseline methods.

###### Strengths ######

--- The algorithm computes the embedding vector of a particular node by using only the local subregion of the network, without requiring the embedding vectors of the remaining nodes.

--- The algorithm outperforms classical baseline methods on small training ratios in the classification task.

###### Weaknesses ######

--- The experimental evaluation is weak and does not allow us to draw meaningful conclusions about the proposed algorithm. The paper does not use most of the recent scalable algorithms, except one, in the empirical analysis.

--- The proposed algorithm does not show significant performance improvement on the link prediction task.

###### Detailed Comments ######

Overall, the paper is well-written and has a well-organized structure. The proposed methodology is interesting, but I have concerns related to various parts of the paper.

In Section 3, the paper starts describing the methodology by firstly introducing the objective function of VERSE and then indicating that it implicitly performs a matrix factorization of the matrix M defined as “log(PPR) + log(n) + log(b)”. In Section 3.1.1, the paper describes the proposed approach as an efficient alternative to factorizing matrix M, preserving the similarities of PPR. Nevertheless, the hashing algorithm is applied to the rows of matrix M, so the approach mainly targets to preserve the similarities in MM^T instead of M as stated in Lemma 3.2. That’s why it seems that Lemma 3.1 does not contribute much to the presentation of the method. The paper also provides the proof of Lemma 3.1 in the Appendix, although this is somehow not relevant to the scope of the paper.

# Theoretical background

--- Lemma 3.2 is very similar to the first part of Lemma 2 in [Weinberger et al., 2009] and Lemma 3.3 relies highly on Theorem 3.2 of the paper [Andersen et al. (2007)]. Thus, it is preferable to provide references for these papers while presenting the lemmas. Similarly, for the proof of Theorem A.5 (ref Theorem 3.4) in the Appendix, the paper should give reference to Theorem 3.2 of Andersen et al. (2007).

--- In the proof of Lemma A.4 (ref Lemma 3.3), I think the expression “... and only if a neighbor of w has positive score” should be replaced with “... and only if there is a node w in the graph such that r[w] greater than \epsilon \times deg(w).” In the proof, the “push” operation is also not defined.

--- In the proof of Theorem 3.5, the symbol \epsilon is used for the InstantEmbedding algorithm corresponding to the term defined for \epsilon-approximate Pagerank vector. However, the \epsilon used for GraphEmbedding indicates the desired precision of the approximation. I think it is not clear that the resulting vectors obtained by these two algorithms with the same epsilon value produce the same vectors. The authors should elaborate more on it in the proof of the theorem. For instance, [Andersen et al. (2007)] demonstrates that the total difference between the PageRank vector and the \epsilon-approximate PageRank vector can be bounded in terms of \epsilon and vol(S) for a given set S.

--- Although the paper mentions that it applies the classic random walk settings in the paper, it actually uses the same updated rules (Algorithm 3 in Appendix) as in Andersen et al. (2007). Thus, the paper works on the PageRank vectors corresponding to the alpha values in the lazy random walk strategy. As the paper states in Section A.7, the alpha for the formulation given in Eq. 1 is different for the same vectors, so Eq. 1 needs to be reformulated.

--- The authors remove the constant “b” but not “n” from the target objective matrix. Is there a particular reason for this choice?

--- In Table 1, the complexity of the proposed approach is provided for learning the representation of a specific node, while the complexities of the baseline methods are given for the whole network structure. I believe that it might be better to introduce an additional parameter for the proposed approach to indicate the desired number of nodes to learn embeddings. For the complexities of the baseline methods, the paper should also provide references for them.


# Experiments

--- For the experiments shown in Figure 1, it seems that the scores for the baseline methods are given for the whole network, while for the proposed method they are stated for only one node. I think it might be better to follow the same strategy for the InstantEmbedding, in order to have a fair comparison -- since we need embeddings for the whole network structure, for instance, for the link prediction experiment in Table 4. The chosen \epsilon values are not also described in the experiments in Figure 1, although \epsilon has a significant influence on the running time.

--- For the link prediction experiment, the paper should use all networks listed in Table 2 in order to have a more complete view of the experimental evaluation of the method.

--- For the classification experiments, although the paper uses 1%, 5%, and 9% training ratios for all networks, it considers 10%, 50%, and 90% training ratios for the Blogcatalog dataset. Is there any particular reason for that? Besides, the VERSE algorithm mostly shows better performance for the link prediction task. How is this behavior explained?

--- In Table 3, the performance of node2vec is better than InstantEmbedding on Blogcatalog network, so its score should be in bold. For the Flickr network, the score of Deepwalk should also be in bold.

--- Another very important aspect has to do with the baselines used in the paper. In the related literature, there are many recently proposed scalable representation learning models that are not used in the experimental evaluation.  Although the paper provides references for some recent scalable algorithms such as RandNE [Zhang et al., 2018] and FREDE [Tsitsulin et al. (2020)], it does not consider them as baseline methods in the experiments. The paper uses solely FastRP [Chen et al., 2019] and the classical node embedding methods to compare the performance of the method, while other recent models are known to perform better than FastRP.  Why recent scalable methods such as NodeSketch [1] and LouvainNE [2], are not considered in the experiments? Besides, why is FREDE (matrix factorization formulation of VERSE)  not used as a baseline?

[1] Dingqi Yang, Paolo Rosso, Bin Li and Philippe Cudre-Mauroux, NodeSketch: Highly-Efficient Graph Embeddings via Recursive Sketching, KDD 2019
[2] Ayan Kumar Bhowmick, Koushik Meneni, Jean-Loup Guillaume, Maximilien Danisch, Bivas Mitra, LouvainNE: Hierarchical Louvain Method for High Quality and Scalable Network Embedding WSDM 2020

# Typos
In Subsection 2.1, “v \in G” should be “v \in V”
In the second paragraph of Subsection B.4.1, “For each methods” should be “For each method”

---

> ### Author Response · Authors · 2020-11-20
> **Response to R4: 1/2**
>
> We thank the reviewer for the careful analysis of our experiments and the suggestions. We believe we provide an extensive experimental treatment of the algorithm given the space constraints of the paper. As suggested, we perform additional experiments using all mentioned baselines and report them below, together with all our answers.
>
>
> ## Theoretical questions
>
> **[$M$ vs. $MM^T$ sketching]**  We refer the argument to FREDE [Tsitsulin et al. (2020)]. They show that matrix factorization-based methods such as NetMF actually preserve the covariance of $M $ instead of the $M$ itself, as they only use the left singular vectors of SVD in the process of embedding generation. These algorithms decompose $M=UΣV^T$, and set an embedding $W=UΣ$. It is easy to see that $MM^T = (UΣV^T)(VΣU^T) = UΣΣU^T$, meaning that setting $W=UΣ$ produces a decomposition of the covariance matrix of $M$. Therefore, Lemma 3.1 has a direct relationship with Lemma 3.2. We will clarify this relationship in the revision.
>
> **[Theorem 3.5]** In both local (InstantEmbedding) and global (GraphEmbedding) procedures, $\epsilon$ has the same meaning, i.e. the desired precision of the PPR approximation. In practice we use the same Push-Flow strategy to generate both local and full graph embeddings.
>
> **[PPR alpha]** Although the classical and lazy random walk settings do determine different $\alpha$ parameterisation, their asymptotic properties are the same. Our choice of parameterisation comes from its ease of representation, but we will change it to consistently reflect our implementation of reference.
>
> **[Constants]** The constant $n$ provides normalisation for the graph size, and is crucial to the performance of our method. We did not find any empirical impact of the constant $b$, and therefore remove it to make the method more simple and easier to analyse.
>
> **[Complexities]** We refer to [Tsitsulin et al. (2018)] for complexities of DeepWalk, node2vec and VERSE. FastRP’s complexity comes directly from its paper. GCN and DGI’s complexity is referenced from [Veličković et al. (2018)].
>
> ## Experimental questions
>
> **[Figure 1]** The reported complexities and the running times/memory usage numbers are all associated with our main defined task: generating a single node embedding. As the other methods require to jointly learn relatable node representation at once, they must embed the entire graph. However, for consistency we will also include a full running time report in the revised paper, and below we provide a table with these numbers including the extended baselines. Because the choice of $\epsilon$ is not constant across all datasets, Figure 1 reports only an overview while we offer additional details about the effect of $\epsilon$, per dataset, in the Appendix.
>
> **[Link Prediction]** We remark that the most genuine link prediction task is represented by the CoAuthor dataset, while this task on the other networks is more related to graph reconstruction. After also evaluating this task on Orkut, the non-discriminative results made us explore different, more relevant tasks that would help us draw a clear line between the evaluated methods.
>
> **[Training percentages]** We use 10/50/90% training data for BlogCatalog since it is a small dataset with skewed label distribution. We use the 10/50/90% split on PPI due to its small size as well; the percentages reported in the paper are a typo we will fix.
>
> **[Table 3]** In table 3, we bold our score if no other method is statistically significantly better than ours (according to Welch’s t-test), and also add an asterisk if our score is the best. Only the first case is true for Blogcatalog.

---

> > ### Author Response · Authors · 2020-11-20
> > **Response to R4: 2/2**
> >
> > **[Extended baselines]** We did not use NodeSketch since it searches for optimal hyperparameters on each and every dataset, defeating the purpose of a general embedding suitable for many downstream applications. We did not initially use FREDE as a baseline because of its quadratic runtime, and our focus on scalability. It does not scale to Flickr, and potentially utilizes up to 560Gb RAM for a dataset which is >50 times smaller than the largest dataset used in our experiments. Regarding random-projection based methods, we initially only considered FastRP in favour of RandNE as being more recent and showcasing better results.
> >
> > We ran all the suggested baselines on the classification task, reporting the Micro-F1 scores and the full running times in the tables below. For all methods we use their suggested default configuration. We evaluate all methods using the same logistic regression model, with two exceptions: for NodeSketch we use an SVC classifier with hamming kernel due to its particular distance preservation properties, while for FREDE we use the originally reported scores, as they employ the same evaluation framework as ours. We note that although FREDE produces better scores, its extremely high running times takes this approach out of the original scope of our paper. In terms of running time required to produce a full graph embedding, although RandNE and LouvainNE are faster on several datasets, their performance is considerably below that of InstantEmbedding. Also, our method running time is directly proportional to the percentage of the graph is required to be embedded.
> >
> > **Micro-F1 Scores:**
> >
> > | Method/Dataset   | PPI    | BlogCatalog | CoCit  | Flickr   | YouTube | Orkut  |
> > | :--------------- | :----- | :---------- | :----- | :------- | :------ | :----- |
> > | DeepWalk         | 16\.08 | 32\.48      | 37\.44 | 31\.22   | 38\.69  | 87\.67 |
> > | node2vec         | 15\.03 | 33\.67      | 38\.35 | 29\.80   | 36\.02  | OOM    |
> > | VERSE            | 12\.59 | 24\.64      | 38\.22 | 25\.22   | 36\.74  | 81\.52 |
> > | FastRP           | 15\.74 | 33\.54      | 26\.03 | 29\.85   | 22\.83  | OOM    |
> > | FREDE            | 19\.56 | 35\.69      | 42\.46 | TimedOut | OOM     | OOM    |
> > | RandNE           | 15\.77 | 32\.79      | 16\.23 | 29\.19   | 24\.57  | 59\.01 |
> > | LouvainNE        | 14\.73 | 22\.28      | 35\.14 | 26\.37   | 33\.52  | 45\.08 |
> > | NodeSketch       | 12\.03 | 29\.65      | 16\.67 | 23\.07   | 32\.02  | OOM    |
> > | InstantEmbedding | 17\.67 | 33\.36      | 39\.95 | 30\.43   | 40\.04  | 76\.83 |
> >
> > **Full running times:**
> >
> > | Method/Dataset   | PPI    | BlogCatalog | CoCit  | Flickr   | YouTube | Orkut   |
> > | :--------------- | :----- | :---------- | :----- | :------- | :------ | :------ |
> > | DeepWalk         | 254    | 711         | 2,767  | 6,035    | 81,168  | 219,003 |
> > | node2vec         | 24\.82 | 67\.8       | 280    | 802      | 7,600   | OOM     |
> > | VERSE            | 87\.5  | 198         | 904    | 1,863    | 31,101  | 84,468  |
> > | FastRP           | 1\.81  | 5\.62       | 7\.21  | 79\.8    | 85\.5   | OOM     |
> > | FREDE            | 150    | 1,194       | 10,954 | TimedOut | OOM     | OOM     |
> > | RandNE           | 2\.05  | 3\.93       | 3\.86  | 39\.4    | 32\.1   | 773     |
> > | LouvainNE        | 0\.62  | 2\.47       | 7\.60  | 17\.70   | 195\.6  | 733     |
> > | NodeSketch       | 19\.9  | 609\.6      | 496\.4 | 3,350    | 2,798   | OOM     |
> > | InstantEmbedding | 0\.27  | 3\.98       | 5\.32  | 53\.6    | 1,254   | 58,070  |
> >
> > In terms of explaining the performance of other methods, VERSE shows consistently good results on the link prediction task [1]. We attribute that fact to better statistical normalisation than plain negative sampling that allows it to distinguish high-degree nodes, which is crucial for link prediction.
> >
> > [1] Benchmarking network embedding models for link prediction: are we making progress? Mara et al., DSAA 2020

---

### Official Review · AnonReviewer1 · 2020-10-29
**Review for InstantEmbedding**

**Rating:** 6
**Confidence:** 3

**Review:**

(Summary)

Unsupervised learning of vector representations for a graph is an important problem for various downstream applications. The authors propose an InstantEmbedding approach that learns d-dimensional embedding for each node in sublinear time. As the paper claims that this approach is a local embedding but globally consistent, users can quickly learn vector representations of nodes based on the local structure, being free from possibly growing the rest of the graph.

Two properties are first defined as qualifications to be a useful local embedding. Locality requires learning algorithms to use only local information. If an approach requires an access to the embeddings of every node in local neighbors or to the entire adjacency matrix, locality is not qualified. Global consistency means that the locally-computed embeddings remain identical even if we compute the embeddings of the entire nodes simultaneously.



(Originality and Contributions)

The proposed algorithm: InstantEmbedding is simple. Once we compute Personalized PageRank (PPR), the algorithm tries to decompose the dense similarity matrix log(PPR) + log(# of nodes) efficiently via two hash functions. The algorithm is designed in a way that global learning is equivalent to local update, automatically satisfying the global consistency. Thus most of the theoretical contributions are based on the idea of implicit factorization by Tsitsulin et al (2020) (which is essentially similar to Levy et al (2014)), the idea of hashing by Weinberger et al (2009), and the sparse computation of PPR by Andersen et al (2007). Instead, this paper conducts extensive study of empirical performance with increasing the error parameter $\epsilon$. The experimental results demonstrate that InstantEmbedding produces competitive embeddings with other methods with a few thousand times better in time and space complexity.



(Questions, Concerns, and Suggestions)

1) Global consistency was expected to be a theoretical property, as given in Section 2.3, such as distance/geodesic preservation between the graph space and the vector space. However, it ended up being chosen more as an algorithmic property, which the proposed algorithm is uniquely achievable by the design. Have you actually measured distortion in various datasets, being able to claim low-distortion?


2) As studied in GEMSEC by Rozemberczki et al (2018), quality embeddings should promote clustering of the nodes (i.e., underlying community structures) as well as preserving the proximities. Considering that one can easily acquire local clusters based only on the local structures of the graph, it will be useful for readers to know whether or not InstantEmbedding can also encode local community structures. And, that could be a significant part of the Locality.


3) Note that Graph Convolutional Convolutional networks can also be trained as an unsupervised fashion and sometimes achieve great generalization without node features. See how synthetic graph is generated featurelessly in GAP by Nazi et al (2019). More importantly, the message passing scheme in GCN is scalable to multi-graph or even hyper-graph, which is not readily clear in the proposed method. Learning hierarchical representation is also questionable.


4) Can you report the degree of scale-free and small-world for each of the graph in experiment? It is an important summary information for distinguishing individual graphs. Hope various graphs used in the experiment have explored a wide range of scale-free and small-world structures.


5) If we carefully read Andersen et al (2007), they did not adopt standard transition matrix given in the proposed model. Asymptotically it would not cause much difference, but it is less clear what’s the impact on the F1-performance given $\epsilon$ tolerance. Note again that InstantEmbedding becomes useful only if the tolerance parameter reaches at a certain level.

---

> ### Author Response · Authors · 2020-11-20
> **Response to R1**
>
> We thank the reviewer for the detailed questions and comments, in particular, their breadth.
>
> In general, while our approach is indeed simple and the techniques we employ are available in the literature, it is competitive with state-of-the-art neural embedding methods while being more scalable and more algorithmically attractive. We emphasise that the other scalable methods (such as FastRP and RandNE; we also added LouvainNE and NodeSketch as suggested by R4) underperform compared to the neural networks and our approach.
>
> We now address the raised points.
>
> [1] There is a difference between “graph embedding” in the sense of embedding a given metric into L2, such as Bourgain (1985) embeddings, and “graph embedding” in a more modern usage of designing both the metric and the embedding to accommodate various graph mining tasks. Our work belongs in the second class: we propose to use a particular metric - log(PPR) - that can be computed locally, and to embed it into L2 via the local hashing. While we can measure distortion of the embedding process, we believe that it would not be a meaningful comparison, as our true focus is on being able to efficiently solve downstream problems.
>
> [2] Thank you for the suggestion. We believe that InstantEmbedding is a path towards formally connecting local clustering with practical graph embedding algorithms. A future work could inspect whether theoretical guarantees could be provided regarding graph cluster preservation properties of PPR-derived embeddings.
>
> [3] While GCNs can be applied on graphs with no features through generation of simple statistical features like the node degree in Nazi et al. (2019), Duong et al. (2019) showed that graph embeddings like DeepWalk or node2vec overwelmingly outperform GCNs when only statistical features are present.
>
>
> [4] To better understand the variety of our chosen datasets, we report the scale-free characteristic of the graphs by fitting a power-low distribution to node degrees, and comparing it to other 2 distributions (exponential and log-normal) through the Kolmogorov-Smirnov test (R = log-likelihood ratio, p = test confidence). We note that on the CoAuthorship, Flickr, YouTube, Amazon2M and Orkut networks, the h0 hypothesis can be rejected (p < 0.05) and thus conclude that the log-normal distribution is a better fit (graphs are not scale-free).
> As an estimator with low variance cannot be provided for the small-world coefficient on larger graphs, we only report two related measures: the pseudo-diameter of the graphs and their global clustering coefficient (transitivity). We observe that the graphs we use in the study are diverse, covering the spectrum of small-world and large-diameter networks, and networks with clear scale-free properties and ones without such properties.
>
> | Dataset     | Exponential Distribution |       | Log-normal Distribution |       | Pseudo-diameter | Transitivity
> | :---------- | :---------------------------------- | :---- | :--------------------------------- | :---- | :------------- | :----------- |
> |             | **R**                                   | **p**     |**R**                                  | **p**     |                |              |
> | PPI         | 12\.49                              | 0\.02 | -0\.56                             | 0\.45 | 8              | 0\.09        |
> | Blogcatalog | 63\.39                              | 0     | -1\.9                              | 0\.16 | 5              | 0\.09        |
> | CoCit       | 354\.09                             | 0     | -2\.61                             | 0\.19 | 25             | 0\.08        |
> | CoAut       | 502                                 | 0     | -102                               | 0     | 30             | 0\.28        |
> | Flickr      | -61\.31                             | 0\.28 | -504                               | 0     | 6              | 0\.18        |
> | YouTube     | 60175                               | 0     | -142                               | 0     | 24             | 0\.006       |
> | Amazon2M    | 41233                               | 0     | -190                               | 0     | 29             | 0\.13        |
> | Orkut       | 33757                               | 0     | -737                               | 0     | 10             | 0\.04        |
>
> [5] While the modified transition matrix in Andersen et al. (2007) changes the definition of PPR, there exists a direct mapping between the Andersen et al. definition and the original one, that we also mention in the Appendix, section A7.
>
> References
> ---
> J. Bourgain “On Lipschitz embedding of finite metric spaces in Hilbert space”, 1985.
> Duong et al. “On Node Features for Graph Neural Networks”, 2019.

---

### Decision · Program_Chairs · 2021-01-07
**Final Decision**

**Decision:**

Reject

**Comment:**

This paper proposes an efficient algorithm to obtain a node embedding based on its local PageRank scores. The proposed approach uses a hashing technique and a local partition approach to make the method more efficient and effective. However, the paper has significant drawback and can be further improved in the following aspects:
1. The experimental evaluation is weak and does not allow us to draw meaningful conclusions about the proposed algorithm.
2. The proposed algorithm does not show significant performance improvement on the link prediction task.